# A redox switch regulates the assembly and anti-CRISPR activity of AcrIIC1

Yanan Zhao[1,5], Jiaojiao Hu[2,5], Shan-Shan Yang[3,5], Jing Zhong[1], Jianping Liu [ID][2], Shuo Wang [ID][4], Yuzhuo Jiao[1], Fang Jiang[1], Ruiyang Zhai[1], Bingnan Ren[1], Hua Cong[1], Yuwei Zhu[4], Fengtong Han[1], Jixian Zhang[1], Yue Xu[1], Zhiwei Huang [ID][4], Shengnan Zhang[2] & Fan Yang [ID][1] ✉

Anti-CRISPRs (Acrs) are natural inhibitors of bacteria's CRISPR-Cas systems, and have been developed as a safeguard to reduce the off-target effects of CRISPR gene-editing technology. Acrs can directly bind to CRISPR-Cas complexes and inhibit their activities. However, whether this process is under regulation in diverse eukaryotic cellular environments is poorly understood. In this work, we report the discovery of a redox switch for *Nme*AcrIIC1, which regulates *Nme*AcrIIC1's monomer-dimer interconversion and inhibitory activity on Cas9. Further structural studies reveal that a pair of conserved cysteines mediates the formation of inactive *Nme*AcrIIC1 dimer and directs the redox cycle. The redox switch also applies to the other two AcrIIC1 orthologs. Moreover, by replacing the redox-sensitive cysteines, we generated a robust AcrIIC1 variant that maintains potent inhibitory activity under various redox conditions. Our results reveal a redox-dependent regulation mechanism of Acr, and shed light on the design of superior Acr for CRISPR-Cas systems.

Over the past decade, the Clustered Regularly Interspaced Short Palindromic Repeat-associated nuclease Cas (CRISPR-Cas) systems have emerged as the most preferred genome-engineering tool in living eukaryotic cells, in both research and therapeutic applications. One of the many advantages that CRISPR-Cas offers over other genome-editing techniques, such as zinc finger nucleases (ZFNs) and transcription activator-like effector nucleases (TALENs), is its ease of programmability[1]. Yet, in addition to programmability, reliability is a greater consideration when evaluating genome-editing tools, especially for applications in human cells. This is because all genome-editing tools, including CRISPR-Cas9, bear potentially deleterious cleavages of non-targeted sites (off-target effects)[2,3]. Against this background, anti-CRISPR (Acr) proteins have drawn much attention ever since their identification, due to their potential applications in precise control of gene editing and reduction of off-target effects of the CRISPR technology[4-7].

Acrs are typically small proteins (~80–150 aa) evolved by phage as a self-defense mechanism against the CRISPR-Cas systems. They can inactivate the CRISPR-Cas systems by inhibiting target DNA binding or cleavage, or in rare cases by preventing proper CRISPR-Cas complex assembly, like type II inhibitor AcrIIC2[6,8–12]. Several Acrs have shown prominent activities of regulating gene-editing events in different cell types, such as two *Spy*Cas9 inhibitors AcrIIA2 and AcrIIA4, and two *Nme*Cas9 inhibitors AcrIIC1 and AcrIIC3[6,8,13]. Among the different Acr proteins characterized so far, AcrIIC1, a broad-spectrum inhibitor against diverse type II-C Cas9 orthologs, is one of the most promising engineering candidates to target the desired Cas9[8,14]. Structural studies show that AcrIIC1 can directly bind to the conserved HNH nuclease domain of Cas9 to abolish its DNA cleavage activity[14]. Recently, a

[1]School of Life Science and Technology, Harbin Institute of Technology, Harbin 150080, China. [2]Interdisciplinary Research Center on Biology and Chemistry, Shanghai institute of Organic Chemistry, Chinese Academy of Sciences, Shanghai 201210, China. [3]State Key Laboratory of Urban Water Resource and Environment, School of Environment, Harbin Institute of Technology, Harbin 150090, China. [4]HIT Center for Life Sciences, School of Life Science and Technology, Harbin Institute of Technology, Harbin 150080, China. [5]These authors contributed equally: Yanan Zhao, Jiaojiao Hu, Shan-Shan Yang. ✉e-mail: fanyang115@hit.edu.cn

designer AcrIIC1 with an improved inhibition potency was successfully employed as a switch to modulate the activity of a hepatocyte-specific *Sau*Cas9 through the Cas-ON approach[15,16]. Therefore, a comprehensive understanding of the regulation mechanism of Acrs for binding and inhibiting Cas9 is essential for advancing our fundamental knowledge of CRISPR biology, and for exploiting naturally evolved inhibitors of CRISPR-Cas technology.

In this work, we find that, in addition to a well-characterized monomeric species, AcrIIC1 from *Neisseria meningitidis* (*Nme*AcrIIC1) can assemble into an inactive dimer under oxidizing conditions. The inactive dimer can reversibly convert into the active monomer under reducing conditions. A further mechanistic study demonstrated that *Nme*AcrIIC1 employs two conserved cysteines to form two pairs of intermolecular disulfide bonds to mediate the dimer's assembly. This enables a redox switch that also applies to two phylogenetically related AcrIIC1 orthologs we tested, indicative of the redox switch's biological significance. More importantly, with the knowledge of the structural and chemical basis of the redox switch of wild-type *Nme*AcrIIC1, we successfully generated an AcrIIC1 mutant with its inactive oxidized state suppressed. This mutant AcrIIC1 may serve as a stable and robust inhibitor of Cas9, resistant to redox fluctuation in various cellular environments, and is beneficial for CRISPR biotechnology.

## Results

### *Nme*AcrIIC1 assembles into a homodimer triggered by oxidization

To investigate the structure and activity of *Nme*AcrIIC1, we first purified the *Nme*AcrIIC1 protein overexpressed in *Escherichia coli* (*E. coli*). Under reducing conditions, the size-exclusion chromatography (SEC) profile shows a single peak at ~10 kDa corresponding to the monomeric species, consistent with the previous studies[14]. Surprisingly, we observed that, in

the absence of a reductant in the purification buffer, *Nme*AcrIIC1 proteins eluted as two separate peaks with apparent molecular weight at 9.6 and 19.3 kDa (Supplementary Fig. 1a, b) on an SEC column. In contrast, SDS-PAGE gel shows a single band at ~10 kDa for both peaks, the correct molecular weight for the *Nme*AcrIIC1 monomer (Supplementary Fig. 1c). Further, [1]H NMR spectra show that the two species are highly similar but not identical (Supplementary Fig. 1d), supporting dimer and monomer, respectively, with similar conformation.

Intriguingly, we noticed that the proportion of dimeric species dramatically increased when the protein sample was kept longer under oxidizing conditions before loading onto the SEC column. This behavior of *Nme*AcrIIC1 is highly reminiscent of proteins with redox states: the oxidized state gradually dominates over the time course of exposure to the air. To examine whether the interconversion between the dimer and monomer of *Nme*AcrIIC1 is under redox regulation, we added excess dithiothreitol (DTT) to the dimer eluent and excess hydrogen peroxide ($H_2O_2$) to the monomer eluent, respectively, and examined the change of their retention positions on the SEC column. In line with our hypothesis, the dimer converted into a monomer by reduction of excess DTT, and the monomer converted into a dimer by oxidization of $H_2O_2$ (Fig. 1a). We further performed native PAGE gel and Ellman's assay to assess the oxidized states of *Nme*AcrIIC1 induced in two ways, either by the air or by the oxidant such as $H_2O_2$. As shown in Supplementary Fig. 2a, the result of native PAGE shows a band with the same size at less than 25 kDa for the oxidized *Nme*AcrIIC1 induced in either way, in comparison with the reduced band at less than 15 kDa. In addition, Ellman's assay result demonstrates no residual -SH groups for the oxidized *Nme*AcrIIC1 induced in, either way, provided an accurate amount of -SH for the reduced *Nme*AcrIIC1 can be calculated against the standard curve fitted with gradient concentrations of DTT mixed with excess DTNB (Supplementary Fig. 2b). Collectively, the

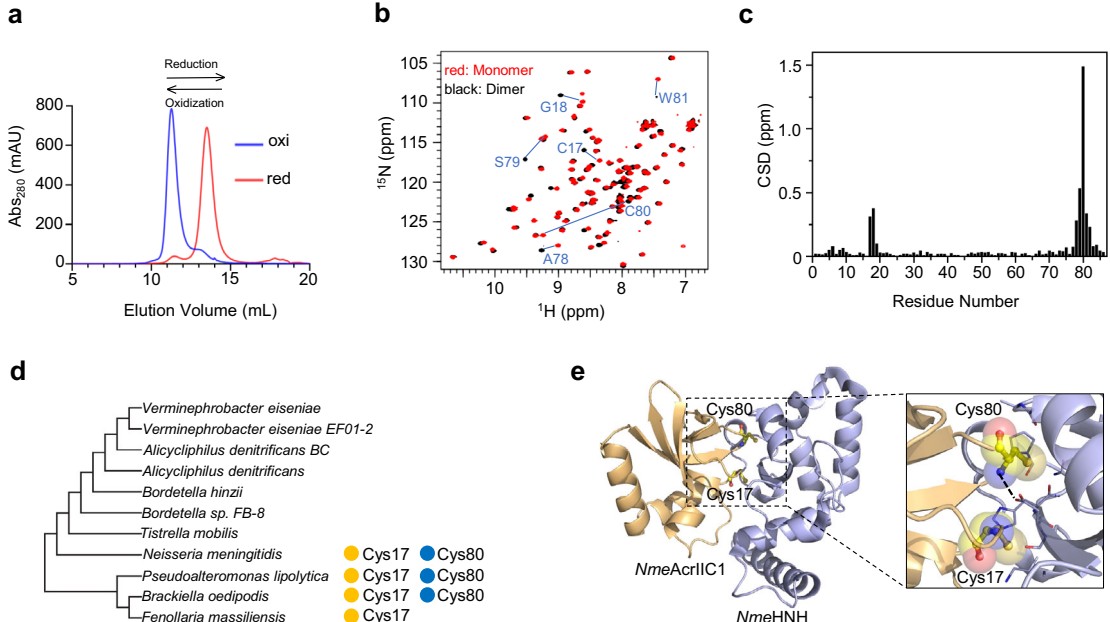

**Fig. 1 | *Nme*AcrIIC1 can be oxidized into a dimer mediated by two conserved cysteines. a** SEC runs show that the retention volume of *Nme*AcrIIC1 changes under different redox conditions. When oxidized, *Nme*AcrIIC1's apparent molecular weight becomes twice as the reduced one, corresponding to a dimer (Supplementary Fig. 1a). **b** Overlay of [1]H-[15]N HSQC spectra for the oxidized dimer and the reduced monomer of *Nme*AcrIIC1. The peaks from the monomer and the dimer are colored red and black, respectively. Residues with significant chemical shift difference (CSD) between the two states are indicated with blue lines and labeled. **c** The bar diagram representing the composite CSDs ($\triangle\delta = [(\delta_{HN})^2 + (\delta_N/6.51)^2]^{1/2}$)

of the HN cross-peaks between the two redox states of *Nme*AcrIIC1, as a function of the residue number. **d** Cys17 and Cys80 of *Nme*AcrIIC1 are present across three related orthologs in the unscaled phylogenetic tree. Cys17 is present in one more related species. A full sequence alignment can be found in Supplementary Fig. 1e. **e** The location of Cys17 and Cys80 of *Nme*AcrIIC1 in its complex with *Nme*HNH (PDB entry 5VGB) is focused with a dotted square. A close-up view of this region with the side chains of Cys17 and Cys80 highlighted with spheres is shown on the right. Source data are provided as a Source Data file.

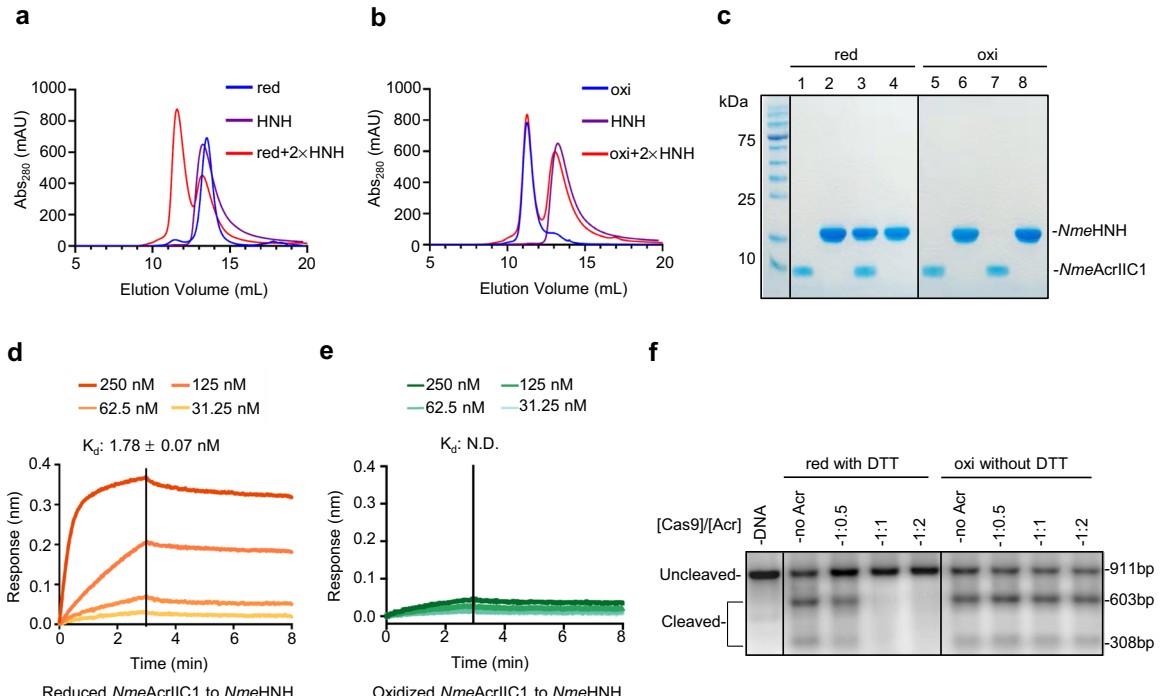

**Fig. 2 | The oxidized *Nme*AcrIIC1 loses its inhibitory activity against Cas9.**
**a** Elution profiles of the SEC binding assay for the reduced *Nme*AcrIIC1 (blue),
*Nme*HNH (purple), or the two incubated together with twofold excess *Nme*HNH
(red). The concentration for each trace was normalized to 0.1 mM of a single pro-
tein. **b** Elution profiles of the SEC binding assay for the oxidized *Nme*AcrIIC1 (blue),
*Nme*HNH (purple), or the two incubated together with twofold excess HNH (red).
**c** Fractions from the assays shown in **a**, **b** were visualized on a 12% SDS-PAGE gel.
Lane 1 to 4 corresponds to reduced *Nme*AcrIIC1, *Nme*HNH, the complex fraction
from the mixture of reduced *Nme*AcrIIC1 with twofold excess *Nme*HNH, and the
residual *Nme*HNH from the mixture above. Lane 5 to 8 corresponds to oxidized

*Nme*AcrIIC1, *Nme*HNH, the earlier eluent from the mixture of oxidized *Nme*AcrIIC1
with twofold excess *Nme*HNH, and the latter eluent from the mixture above.
**d, e** Binding affinities determined by BLI for the reduced *Nme*AcrIIC1 and the oxi-
dized *Nme*AcrIIC1 to *Nme*HNH, respectively. The reduced *Nme*AcrIIC1 shows a $K_d$ of
1.78 nM, while the oxidized *Nme*AcrIIC1 gives a very weak response <0.05, indi-
cating very poor binding. **f** DNA cleavage assays were conducted in the presence of
the reduced *Nme*AcrIIC1 with five-fold excess DTT (relative to Acr) in the buffer, and
the oxidized *Nme*AcrIIC1 without DTT, respectively. Data were replicated inde-
pendently more than three times with similar results and the representative one is
shown. Source data are provided as a Source Data file.

oxidized forms of *Nme*AcrIIC1 induced in two ways are identical in
terms of their oligomeric and redox states. To be consistent, for the
following assays, we made the oxidized *Nme*AcrIIC1 by using $H_2O_2$,
which takes a shorter incubation time as long as an appropriate
amount of oxidant is used.

We next characterized the two redox forms of *Nme*AcrIIC1 by
using NMR spectrometry. We prepared the $^{15}N$ and $^{13}C$ labeled protein
samples of monomer and dimer, collected 2D $^1H$–$^{15}N$ HSQCs, and
assigned the backbone H-N cross-peaks using standard triple reso-
nance procedures for each of them. As shown in Fig. 1b, the $^1H$-$^{15}N$
HSQC counts about 80 sharp and well-dispersed peaks with almost
even intensities for the monomer. Strikingly, the $^1H$-$^{15}N$ HSQC spec-
trum for the dimer also shows a set of ~80 cross-peaks, indicative of a
symmetric structure, at least when averaged over the NMR time scale.
Moreover, the $^1H$-$^{15}N$ HSQC spectrum of the monomer spontaneously
turned into that of the oxidized dimer induced with $H_2O_2$, further
confirming that the oxidized *Nme*AcrIIC1 induced by different ways are
identical from a structural perspective (Supplementary Fig. 2c).

Notably, the two HSQCs of *Nme*AcrIIC1 monomer and dimer
overlap quite well. Only less than ten residues sequentially adjacent to
Cys17 and Cys80 exhibit significant large chemical shift deviation
(CSD) (Fig. 1c), which strongly suggests that Cys17 and Cys80 could be
the key residues triggering the redox state transition. Importantly,
phylogenetic analysis of 11 AcrIIC1 orthologs from different bacteria
originally annotated by Pawluk et al[8]. shows that Cys17 and Cys80 in
*Nme* are conserved across at least three related species (Fig. 1d and
Supplementary Fig. 1e), further supporting their functional roles from
an evolutionary standpoint. In addition, by carefully examining the
structure of *Nme*AcrIIC1 bound to the HNH nuclease reported

previously[14] (Fig. 1e), we noticed that Cys17 and Cys80 are both located
on the protein surface. More importantly, both cysteines are involved
in the interface of *Nme*AcrIIC1's complex with HNH nuclease and play a
role in the interaction. Since our study shows that Cys17 and Cys80 are
also involved in dimer formation, we hypothesized that dimeric
*Nme*AcrIIC1 would be unable to bind Cas9.

## The oxidized dimer of *Nme*AcrIIC1 cannot bind to HNH and therefore loses its inhibitory activity against Cas9

Structural work demonstrated that AcrIIC1 functions through binding
to and conformationally restraining the HNH nuclease of Cas9[14]. We
thus examined whether the oxidized dimer of *Nme*AcrIIC1 can bind to
HNH nuclease and exert inhibitory activity against Cas9 like its
reduced monomeric form. We characterized the interaction between
*Nme*AcrIIC1 and *Nme*HNH with SEC and measured the equilibrium
binding affinity by Bio-Layer Interferometry (BLI). We first incubated
the reduced *Nme*AcrIIC1 with twofold excess *Nme*HNH and fractio-
nated the mixture over a Superdex 75 SEC column. A significant shift of
*Nme*AcrIIC1 was observed due to the binding of *Nme*AcrIIC1 to
*Nme*HNH (Fig. 2a, c). In sharp contrast, incubated with two fold excess
*Nme*HNH, the oxidized *Nme*AcrIIC1 showed no shift, indicative of no
binding with *Nme*HNH, at least at µM concentration used for SEC
(Fig. 2b, c). To further measure the detailed binding kinetics for both
states of *Nme*AcrIIC1 to *Nme*HNH, we performed BLI to obtain the
equilibrium dissociation constant $K_d$. Fitting of the data yielded a $K_d$
value of 1.8 nM for the reduced *Nme*AcrIIC1(Fig. 2d), comparable to
6.3 nM previously measured by Isothermal Titration Calorimetry
(ITC)[14]. Yet, for the oxidized *Nme*AcrIIC1, we were unable to measure
any appreciable affinity (Fig. 2e).

Additionally, we performed the NMR titration experiments to explore the interactions of different redox forms of *Nme*AcrIIC1 with *Nme*HNH at an atomic level. Consistent with previous results, NMR titration of the reduced *Nme*AcrIIC1 with *Nme*HNH yielded another set of cross-peaks being in slow conformational exchange with the original peaks, indicative of strong interaction between the titrate and the titrant with a $K_d$ falling into nM-μM range, shown in Supplementary Fig. 3a. In sharp comparison, the titration of the oxidized *Nme*AcrIIC1 with *Nme*HNH didn't change the cross-peaks even at a half mM concentration (Supplementary Fig. 3b).

Given the loss of the interaction of the oxidized *Nme*AcrIIC1 to *Nme*Cas9, we next sought to examine whether the oxidized *Nme*AcrIIC1 dimer exhibits inhibitory activity against *Nme*Cas9. We conducted well-established DNA cleavage assays in the presence of an Acr protein in vitro. We set up the DNA cleavage assay with the reduced and the oxidized *Nme*AcrIIC1 under reducing and non-reducing conditions, respectively (Fig. 2f). Under the reducing condition, with DTT (five-fold excess relative to Acr) maintained in the reaction buffer, the reduced *Nme*AcrIIC1 was able to inhibit the DNA cleavage by *Nme*Cas9 completely with only a 1:1 ratio of Cas9 to Acr. In contrast, under the non-reducing condition containing no reductant in the reaction buffer, the oxidized *Nme*AcrIIC1 showed no impact on *Nme*Cas9's function at all, even at a higher Acr ratio of 1:2. This result demonstrates that the oxidized dimer of *Nme*AcrIIC1 represents its inactive state without Cas9 inhibitory activity.

### Structures of the two redox states of *Nme*AcrIIC1

To further delineate the structural basis underlying the formation of the inactive dimeric state of *Nme*AcrIIC1, we set out to determine the structures of both redox states of *Nme*AcrIIC1. We first performed solution NMR to determine the structure of the monomeric reduced *Nme*AcrIIC1. With larger than 90% of the chemical shifts for the backbone and the side chains assigned, combined with a good number of NOE-derived distance restraints (Supplementary Table 1), we obtained a high-quality ensemble of solution structure for the reduced *Nme*AcrIIC1. The superposition of the 20 lowest-energy conformers together with the ribbon diagram of the representative structure is shown in Fig. 3a, b. The root mean square deviation (RMSD) of the 20 conformers for the backbone of the secondary structural region is only 0.19 Å, indicating an ordered and rigid conformation. Overall, the reduced *Nme*AcrIIC1 adopts a similar conformation with a β1β2β3α1α2β4β5 topology as its holo-form in complex with *Nme*HNH, exhibiting an RMSD of 1.00 Å for the full-length backbone (Supplementary Fig. 4a). Importantly, Cys17 and Cys80 are highly solvent-exposed in the monomeric structure, and their side chains exhibit multiple conformations on NMR time scale (Fig. 3a), in support of our hypothesis that these two cysteines could be redox-active.

For the oxidized dimer, NMR experiments showed that the chemical shifts for $C_β$ of Cys17 and Cys80 shift downfield to 44.3 and 37.1 ppm, respectively, revealing that Cys17 and Cys80 are both involved in the disulfide linkage[17] (Supplementary Fig. 4b). However, our efforts to obtain intermolecular restraints using $^{15}$N, $^{13}$C-filtered/$^{15}$N, $^{13}$C-edited NOESY experiments were unsuccessful. Fortunately, we managed to grow protein crystals formed by the oxidized *Nme*AcrIIC1 and solve the crystal structure of the dimer with a resolution of 1.61 Å by using X-ray crystallography (Supplementary Table 2). As shown in Fig. 3c, in the dimeric structure, one of the protomers is rotated roughly 180 degrees relative to the other partner, which then allows for the dimerization via two intermolecular disulfide bonds. Each subunit of the dimeric form aligns with the reduced monomeric form of *Nme*AcrIIC1 with a small RMSD of 0.68 and 0.73 Å, respectively (Supplementary Fig. 4c, d). Strikingly, the two subunits assemble mainly through two parallel intermolecular disulfide bonds formed by Cys17 and Cys80, but not hydrophobic or electrostatic interaction that is commonly observed in mediating protein–protein interaction. The subunit orientation of the

dimer is restricted to a great extent by the two intermolecular disulfide bonds, shown in Fig. 3c, d. This unique dimerization pattern is critical in regulating the dynamic switches between the active monomer and the inactive dimer in response to different redox conditions, which will be discussed in the next section of the results.

To preclude an artificial structural arrangement caused by crystal packing, we further investigated the behavior of the dimer in solution by NMR. First, in this scenario where redox state exchange is coupled with monomer-dimer conversion, CSD uncovers residues impacted either by thiol-disulfide bond transition or by interfacial contacts to the other subunit. We observed that significant CSDs only occur for the residues next to Cys17 and Cys80 but not to other regions (Fig. 1b, c), which echoes the crystal structure that Cys17 and Cys80 are the key residues involved in the dimer interface. Second, the estimated rotational correlation time ($T_c$) for the oxidized dimer is almost twice as that for the reduced monomer (11.6 versus 6.0 ns), whereas the trends of $^{15}$N backbone relaxation data of $R_1$, $R_2$, and Het-NOE align well between the two states on a residue-by-residue base, which reveals a rigid dimer with restricted orientation for each subunit but lacking large interfacial surface that is supposed to alter the dynamic properties of the residues involved (Fig. 3e). Together, the CSD and $^{15}$N relaxation data for both forms of *Nme*AcrIIC1 in solution strongly support the crystal structure, stating that the crystal structure of the oxidized dimer can reliably represent the dimer's behavior in solution.

Importantly, compared with the structure of *Nme*AcrIIC1's complex with *Nme*HNH, the counterpart subunit of dimeric *Nme*AcrIIC1 occupies the position of the *Nme*HNH, burying the "hot spot" residue Ser79, thereby inhibiting the interaction between *Nme*AcrIIC1 with *Nme*HNH through steric hindrance (Fig. 3f, g). Thus, the dimeric structure rationalizes how oxidization and dimerization of *Nme*AcrIIC1 result in the disruption of *Nme*HNH binding and the loss of inhibitory activity against Cas9.

### Cys17 and Cys80 form the key redox switch to mediate *Nme*AcrIIC1's redox cycle

We next asked whether *Nme*AcrIIC1 can be redox recycled both structurally and functionally via the pair of cysteines as a redox switch. First, we examined the interconversion of the redox structures by using $^1$H-$^{15}$N HSQC. Treated with excess $H_2O_2$ as the oxidant, the structure of the reduced *Nme*AcrIIC1 spontaneously turned into that of the oxidized/dimeric one. We were able to observe the conversion back to the monomeric form upon the addition of excess DTT to overcome the oxidizing effects of $H_2O_2$. This structural interconversion is marked by HN cross-peak shifts of the residues next to Cys17 and Cys80 (Fig. 4a, b). We next explored whether the functional activity of *Nme*AcrIIC1 is coupled with its structure and can be recycled in response to different redox conditions. NMR titration experiments showed $H_2O_2$-processed *Nme*AcrIIC1 loses its binding to *Nme*HNH, while the re-reduced *Nme*AcrIIC1 from the $H_2O_2$-processed one can bind to *Nme*HNH as competent as the nascent one. Furthermore, we performed the DNA cleavage assay to assess the inhibitory activity of *Nme*AcrIIC1 within the redox cycle. The inhibitory activity of the reduced *Nme*AcrIIC1 was completely impaired by the addition of excess $H_2O_2$, and then 100% recovered after the addition of excess DTT (Fig. 4c, d). Thus, our results show that the structure and functional activity of *Nme*AcrIIC1 can be recycled between its different redox states.

At the molecular level, good redox switches should be accessible and have intermediate half-lives and sufficient reactivity[18]. According to these criteria, the pair of Cys17 and Cys80 qualifies as a superior redox switch supported by three lines of evidence: First, as shown in the structures of both redox states of *Nme*AcrIIC1 (Fig. 3a–d), Cys17 and Cys80 are indeed highly accessible in forms of either thiols or disulfide bonds. Second, by examining the electron density map of the oxidized *Nme*AcrIIC1, we observed that both of the intermolecular

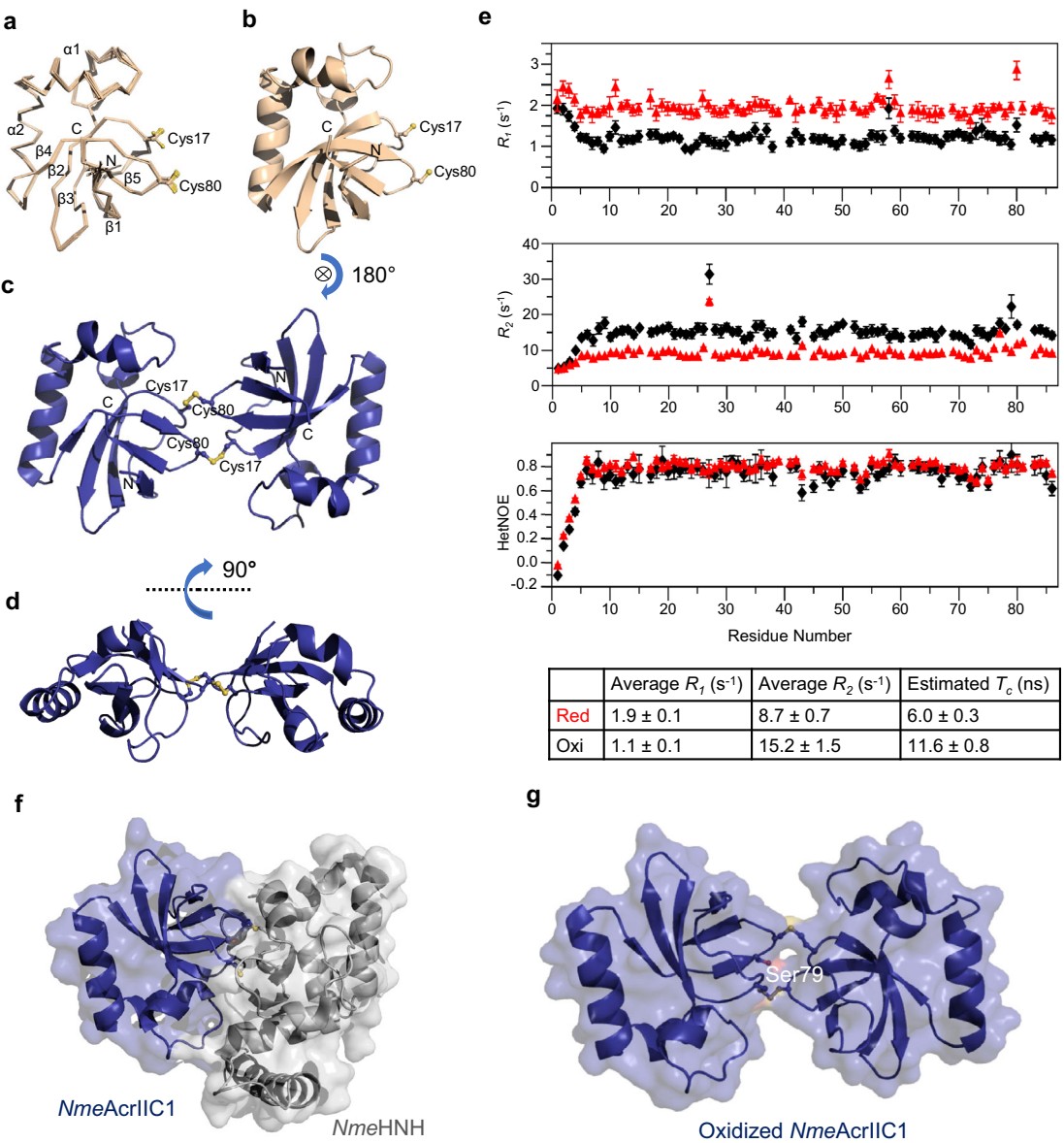

**Fig. 3 | Structures and dynamics for both redox states of *Nme*AcrIIC1.**
**a** Superposition of the 20 lowest-energy structures of the reduced *Nme*AcrIIC1 calculated using solution NMR. Secondary structures are labeled, and the side chains of Cys17 and Cys80 are drawn in sticks. **b** The representative lowest-energy structure of the reduced *Nme*AcrIIC1 is shown in the ribbon diagram. **c** Crystal structure of the dimeric oxidized *Nme*AcrIIC1. **d** The same as **c**, rotated by 90° along the horizontal axis drawn in a dotted line. **e** $^{15}$N backbone relaxation data for the reduced *Nme*AcrIIC1 (red) and the oxidized *Nme*AcrIIC1 (black). $^{15}$N longitudinal relaxation rate constant $R_1$ (top panel), $^{15}$N transverse relaxation rate constant $R_2$ (middle panel), and heteronuclear {$^1$H}-$^{15}$N NOE values (bottom panel) plotted as a function of the residue number. Rotational correlation times $T_c$ were estimated using average $R_1$ and $R_2$ using Eq. (1) and listed in the table below. $n = 2$ independent experiments have been performed. Dots represent mean and error bars represent SD. **f** Surface representation of the complex structure of *Nme*AcrIIC1 bound to *Nme*HNH. **g** Surface representation of the structure for the oxidized *Nme*AcrIIC1. Residue Ser79, crucial to the recognition of *Nme*HNH, is highlighted and labeled. Source data are provided as a Source Data file.

disulfide bonds are not fully occupied, indicative of dynamic short-lived chemical groups (Supplementary Fig. 5a). Third, the pKₐ values of Cys17 and Cys80 determined using pH titration are 8.9 and 8.3, respectively (Supplementary Fig. 5b), shifting from the average value (8.5 ± 0.5) in the opposite direction[19]. This significant pKₐ shifting indicates strong interactions could occur between Cys17 and Cys80 by sharing the proton remaining on the Cys17 after the ionization of Cys80, consequently accelerating the redox reaction[20,21]. Taken together, our results demonstrate that the Cys17-Cys80 pair serves as a highly active redox switch to mediate the redox cycle of *Nme*AcrIIC1 for dynamically regulating its functional activity in response to different redox conditions.

## AcrIIC1 variant with robust anti-Cas9 activity in diverse redox environments

As the oxidative condition can diminish the activity of *Nme*AcrIIC1, we next asked whether we could generate a robust *Nme*AcrIIC1 with potent and constant activity irrespirative to various redox environments. Based on our structural and functional studies, we optimized *Nme*AcrIIC1 with Cys17 and Cys80 replaced to suppress the formation of its inactive oxidative state. In one mutant, the two cysteines are substituted with serine residues due to structural similarity, designated as AcrIIC1-SS. The other mutant uses alanine instead of serine (AcrIIC1-AA) to explore whether a smaller side chain could benefit when interacting with *Nme*HNH. Initial in silico evaluation of their

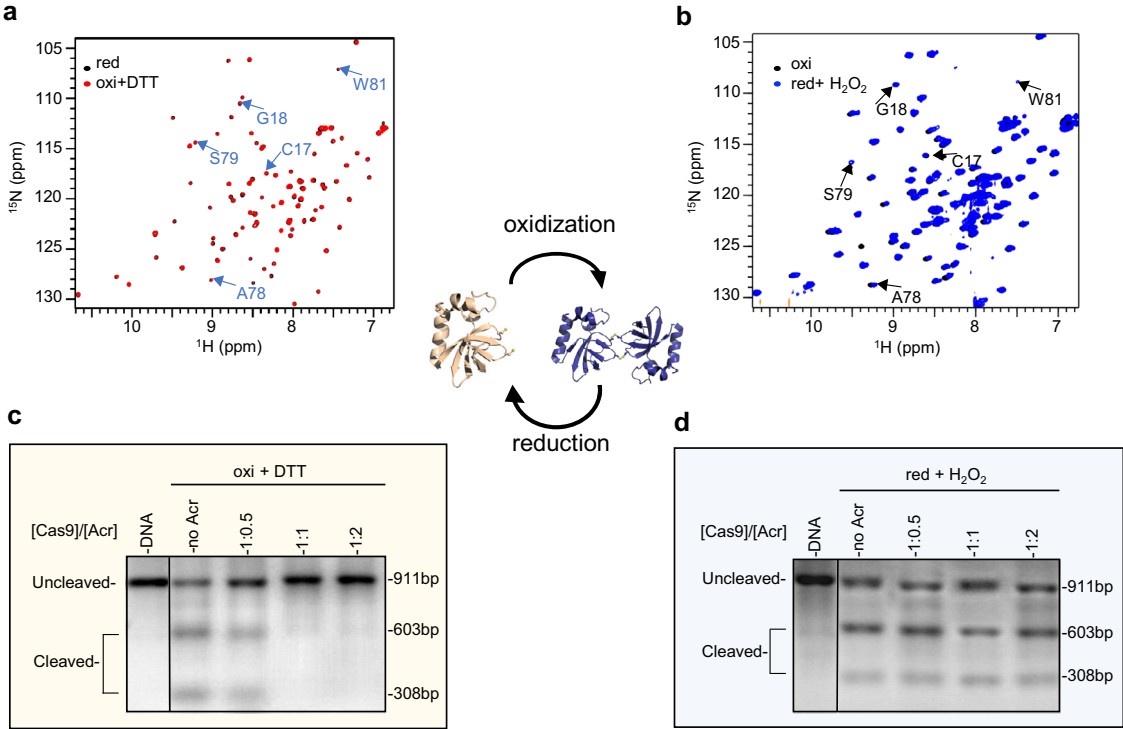

**Fig. 4 | The two redox states of *Nme*AcrIIC1 can be interconverted and mediated by the pair of Cys17-Cys80. a** Superimposition of $^1H$-$^{15}N$ HSQC spectra of the reduced *Nme*AcrIIC1 (black) and that after recycling from the oxidized state (red). To better demonstrate the change of the structure from $^1H$-$^{15}N$ HSQC, peaks with significant CSD between the two states of *Nme*AcrIIC1 are indicated with blue arrows and labeled. **b** Superimposition of $^1H$-$^{15}N$ HSQCs of the oxidized *Nme*AcrIIC1 (black) and that after recycling from the reduced state (blue). **c** The DNA cleavage reaction inhibited by the re-reduced *Nme*AcrIIC1 from its oxidized state. **d** The result of DNA cleavage assay in the presence of an increasing ratio of the $H_2O_2$-oxidized *Nme*AcrIIC1. The DNA cleavage assays have been repeated independently more than three times with similar results. Source data are provided as a Source Data file.

interfacial contacts to *Nme*HNH using Rosetta InterfaceAnalyzer[22] suggested that, the side chain -OH from serine of AcrIIC1-SS would better mimic the -SH from cysteine of the wild-type *Nme*AcrIIC1 and near-completely reserves the hydrogen bonding to *Nme*HNH (Fig. 5a). In addition, the Rosetta outputs favor the AcrIIC1-SS mutant in several aspects, including better shape complementary value, larger interface surface area, etc. Thus, we chose AcrIIC1-SS for the following functional experimental assessment.

To examine the functional behavior of AcrIIC1-SS, we measured AcrIIC1-SS's binding capacity to *Nme*HNH using BLI and NMR titration, and its inhibitory activity against Cas9 using DNA cleavage assay. With or without reductants in the buffer, AcrIIC1-SS exhibits a $K_d$ to *Nme*HNH at 12.7 or 10.8 nM (Fig. 5b and Supplementary Fig. 6a), achieving comparable binding affinity as that of the wild-type AcrIIC1 (wtAcrIIC1). Consistent with this strong interaction, NMR titration and gel filtration result of AcrIIC1-SS with *Nme*HNH revealed a tight binding with a similar slow-exchange pattern as the wtAcrIIC1 (Supplementary Fig. 6b–d). Most importantly, with the redox-sensitive cysteines being replaced, AcrIIC1-SS doesn't undergo the redox cycle as wtAcrIIC1 does, offering better stability over wtAcrIIC1 under various redox conditions (Fig. 5c). As for the activity, under reducing conditions, AcrIIC1-SS inhibits the DNA cleavage activity of Cas9 to a comparable extent as the wtAcrIIC1 (Fig. 5d top panel). Moreover, in sharp contrast to the significant drop of the inhibitory activity of wtAcrIIC1, the activity of AcrIIC1-SS remains constantly high under various non-reducing conditions, either exposed to the atmosphere or excess $H_2O_2$ (Fig. 5d middle and lower panel). Thus, as summarized in Fig. 5e, wtAcrIIC1's activity is prone to vary upon redox potential changes and can undergo inactivation via disulfide-mediated dimerization. Whereas, with the redox-sensitive Cys residues being replaced without compromising its binding capability to Cas9, AcrIIC1-SS would not be affected by redox potential fluctuation, serving as a highly robust AcrIIC1 variant.

To further investigate the redox regulation of wtAcrIIC1 biologically and to examine how robust AcrIIC1-SS behaves in various redox environments in cells, we established an antimicrobial susceptibility assay to qualitatively assess the anti-CRISPR activity of AcrIIC1 in *E. coli*. Taking advantage of the pCas-pSG-mediated constitutive expression of Cas9 and a sgRNA targeting the kanamycin resistance gene[23], we succeeded to observe a larger inhibition ring around the paper disk impregnated with kanamycin on the bacterial lawn of growth relative to the negative control, as shown in Fig. 6a–c panel 1, 2. Very importantly, co-expression of wtAcrIIC1 induced by IPTG restored the inhibition zone to a comparable size where no CRISPR-Cas9 functions (Fig. 6a, c panel 3, 4), strongly indicating that Cas9 is mostly inhibited by the overexpressed wtAcrIIC1. By contrast, the inhibition zone expanded to a similar size as where wtAcrIIC1 was not expressed, when the bacterial cells were exposed to 1 mM $H_2O_2$ during wtAcrIIC1 expression, implying substantial activity loss of wtAcrIIC1 under the oxidative environment (Fig. 6a, c panel 5, 6). Moreover, native PAGE gel confirmed almost complete conversion of wtAcrIIC1 to its oxidized dimeric form in the cell under $H_2O_2$ stimulus (Supplementary Fig. 7a). Not surprisingly, when co-expressed with AcrIIC1-SS, the inhibition zone remained the same size with or without $H_2O_2$ stimulus, as shown in Fig. 6b, c panel 3, 4, 5, 6, stating that AcrIIC1-SS behaves as a superior Cas9 inhibitor with more stable activity in vivo.

## Discussion

Bacterial cells are constantly experiencing redox potential fluctuation, caused by reactive oxygen species (ROS) generated as a product of aerobic metabolism or exogenous stimuli such as UV radiation and oxidative stress from host defense. To adapt to this redox variation,

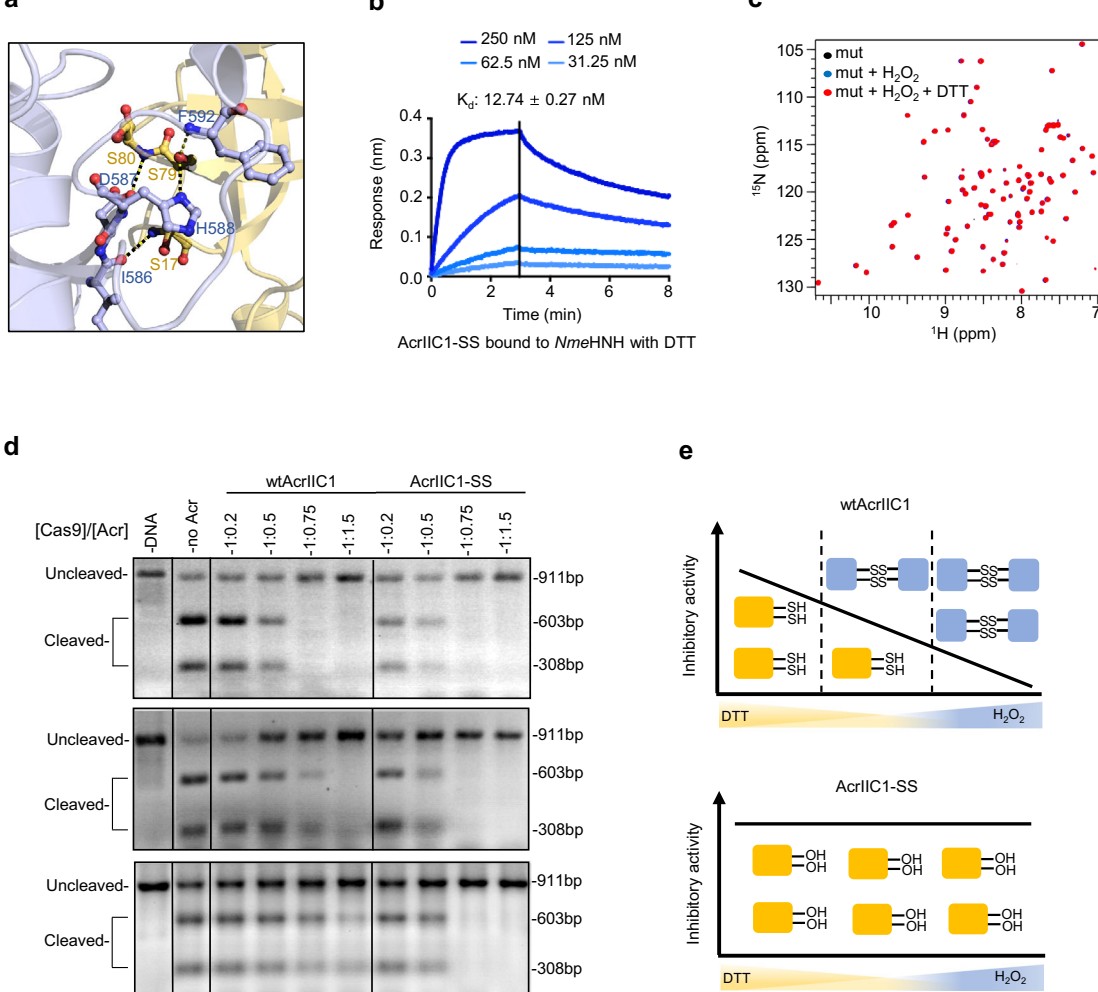

**Fig. 5 | AcrIIC1-SS is a robust AcrIIC1 variant. a** A close-up view of the interface of AcrIIC1-SS in complex with *Nme*HNH. The structural model was built using the SWISS-MODEL website (https://swissmodel.expasy.org) with PDB entry 5VGB as the template. AcrIIC1-SS is colored yellow and *Nme*HNH is colored purple. Key residues involved in interfacial hydrogen bonds are drawn in sticks and labeled. **b** $K_d$ determined by BLI for AcrIIC1-SS bound to *Nme*HNH is 12.7 nM with excess DTT in the reaction buffer. **c** Overlay of $^1$H-$^{15}$N HSQC spectra of AcrIIC1-SS (black), AcrIIC1-SS treated with excess $H_2O_2$ (blue), and AcrIIC1-SS treated with $H_2O_2$ and more DTT in sequence (red) indicate the structure of AcrIIC1-SS is not affected by redox reagents. **d** DNA Cleavage assays were conducted with wtAcrIIC1 and AcrIIC1-SS under different redox conditions. Three reaction conditions were used: the reducing condition with 1 mM DTT in the buffer (top panel), the non-reducing condition without DTT or $H_2O_2$ in the buffer (middle panel), and the oxidizing condition with a fivefold molar ratio of $H_2O_2$ relative to Acr in the buffer (bottom panel). The experiments have been repeated more than three times with similar results. **e** Schematic representation of the inhibitory activities of wtAcrIIC1 (top) versus AcrIIC1-SS (bottom) under different redox environments. Source data are provided as a Source Data file.

bacteria have evolved a comprehensive redox signaling system involving a large number of redox-sensing regulatory proteins[24]. Our study reveals a naturally occurring redox switch of *Nme*AcrIIC1. Moreover, we demonstrate that *Nme*AcrIIC1 is subject to dynamic redox regulation both structurally and functionally mediated by a key cysteine pair, which is conserved in the other two AcrIIC1 orthologs from phylogenetically related species *Brackiella oedipodis* (*Boe*) and *Pseudoalteromonas lipolytica* (*Pli*). Importantly, by using SEC and NMR $^1$H spectra, we demonstrated that these two AcrIIC1 orthologs undergo redox cycles as well. As for *Boe*AcrIIC1, which shares 28.4% of sequence identity with *Nme*AcrIIC1, we observed a similar pattern of monomer-dimer structural transition in response to redox condition change in the buffer (Supplementary Fig. 8a–c). *Pli*AcrIIC1, which contains an extra alpha helix extended at its C-terminus as modeled by AlphaFold[25] (Supplementary Fig. 8d), is highly prone to aggregation under oxidizing conditions. Nevertheless, we observed the transition between the reduced monomer and the oxidized oligomers of *Pli*AcrIIC1 by using SEC and $^1$H NMR spectra (Supplementary Fig. 8e–g). It's very possible that the redox switch identified from AcrIIC1 may serve as a

common regulation mechanism shared by other Acr proteins. It's been only less than ten years since the first anti-CRISPR protein was identified[26], and there is so much to be uncovered in this field, including more members in the AcrIIC1 family and other Acr proteins that might possess redox-sensitive cysteines.

Our in vitro assays demonstrated the redox switch of *Nme*AcrIIC1 and elucidated the structural basis underlying this dynamic regulation process. Recent publications involving experimental studies for AcrIIC1 indicate that previous in vitro assays were performed with reductants maintaining reducing experimental conditions (listed as Supplementary Table 3)[8,12,14,15,27–31]. For example, in the structural study of *Nme*AcrIIC1 complexed with HNH, they kept 0.5 -1 mM TCEP-NaOH in all purification buffers and 1 mM DTT or 1 mM TCEP-NaOH for all in vitro cleavage and binding assays. It could be the reducing experimental condition that precluded the chance of forming the oxidized AcrIIC1. As for previous in vivo assays, since only apparent activity of AcrIIC1 can be observed without characterization of the molecular state, it's hard to tell the exact redox state of AcrIIC1 molecules in the cell. To assess the intracellular state

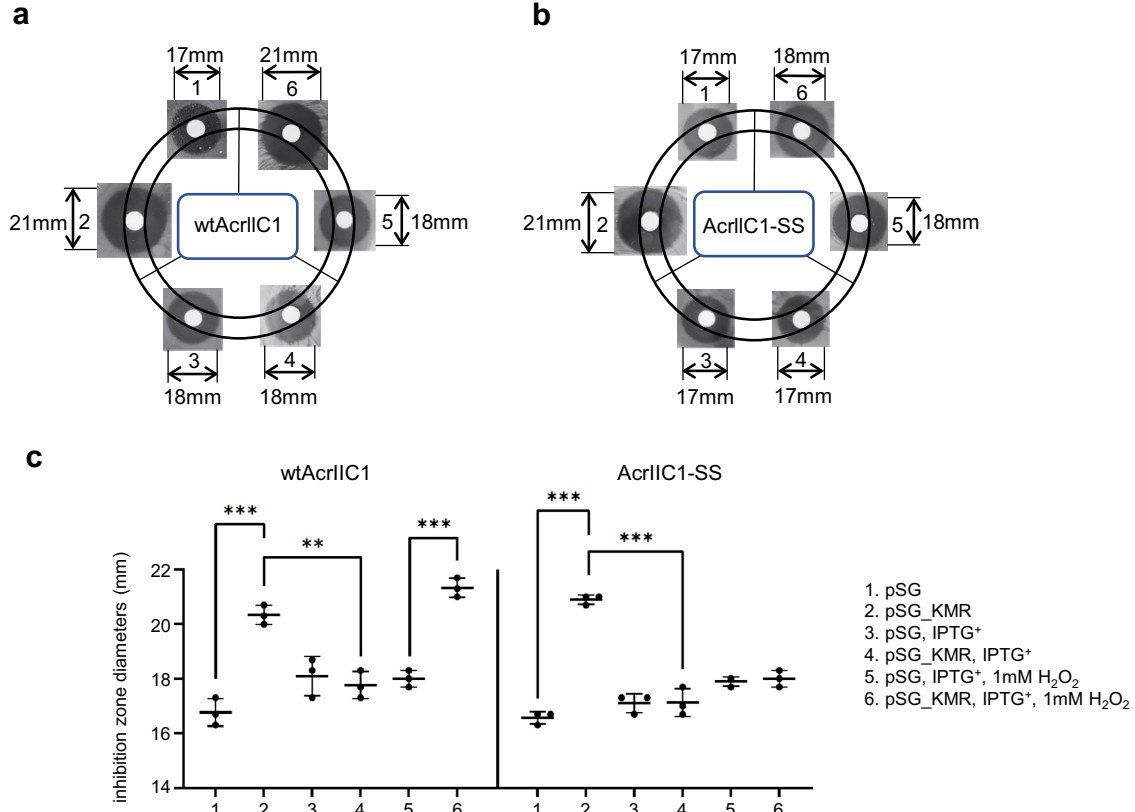

**Fig. 6 | Antimicrobial susceptibility assays accessing wtAcrIIC1 and AcrIIC1-SS activities in vivo. a** Panels 1, 2, assays without wtAcrIIC1 expression: pSG represents negative control carrying no spacer; pSG_KMR represents *kmR* spacer-introduced plasmid. The gene-editing ability of the two-plasmid system pCas-pSG was confirmed by a significant increase in the size of the inhibition ring relative to the negative control, with a representative diameter of 21 mm vs 17 mm. Panels 3, 4, assays with wtAcrIIC1 induced by IPTG. Co-expression of wtAcrIIC1 restored the inhibition ring because it binds to and deactivates Cas9. Panels 5, 6, assays with wtAcrIIC1 induced and 1 mM $H_2O_2$ stimulus. wtAcrIIC1 failed to restore the inhibition ring under $H_2O_2$ stimulus due to the loss of its activity under the oxidative environment. **b** Co-expression of AcrIIC1-SS can restore the inhibition ring under various redox environments because its activity is resistant to redox fluctuation.

Assay panels are in the same order as **a**. **c** Statistical analysis of the inhibition zone diameters from **a**, **b**. The left part shows the size of the inhibition ring varies in response to environmental redox change when wtAcrIIC1 is co-expressed with Cas9. The right panel shows the size of the inhibition ring remains constant and resistant to environmental redox changes when AcrIIC1-SS is co-expressed with Cas9. Lines in the plots indicate means, dots indicate individual data points from three independent experiments, and every dot is calculated by averaging three rings' diameters from one plate. Error bars represent SD. Unpaired two-sided *t*-test, \*\**P* < 0.01, \*\*\**P* < 0.001. The exact *P* values for wtAcrIIC1 panels 1 with 2, 2 with 4, and 5 with 6 are 0.0005, 0.0019, and 0.0002, respectively. The *P* values for AcrIIC1-SS panels 1 with 2 is <0.0001, and panels 2 with 4 is 0.0003. Source data are provided as a Source Data file.

of AcrIIC1, we measured the redox potentials for the reducing buffers we used (listed as Supplementary Table 4), and compared them with the intracellular redox potentials reported previously. It seems that normal cytoplasmic values (−260 to −200 mV) fall within the range of our reducing buffers (−360 to −200 mV)[32,33], indicating most of the AcrIIC1 molecules should be at their reduced state under normal physiological conditions. Under oxidative stress, local redox potential elevation in cells is not easy to measure at this point, due to the lack of a stable redox sensor working in this condition. However, it was reported that cells exposed to a more oxidizing potential (0 mV) had stimulated $H_2O_2$ production[34], which can induce the inactive dimer of AcrIIC1 based on our in vitro and in vivo data. An important question is why do phages add this additional layer of regulation on AcrIIC1? We propose one possible explanation: This inactivation mechanism evolved for AcrIIC1, a broad-spectrum Cas9 inhibitor, could be an effective strategy for phage to fine-tune the level of active Acrs when its host bacterium experiences oxidative stress. Therefore, the phage could protect itself by helping its host to defend against other more lethal phages through CRISPR-Cas systems during stress, achieving a win-win situation with the host. Indeed, it was previously documented that prophages do help bacteria cope with adverse environments, including osmotic, oxidative, and acid stresses[35]. Our work opens a door for investigating the

dynamic regulation of Acrs under the redox switch and leaves more critical questions to be addressed in the future.

As a key member of the Acr toolbox for Cas9-based gene editing, *Nme*AcrIIC1 has a wide range of applications in different cell types and tissues[8,15]. From a biotechnological point of view, we should consider redox inactivation when applying *Nme*AcrIIC1, potentially more other Acrs, in cells with unpredictable redox potential, such as cancer cells often with elevated levels of ROS[36]. Recently, David Liu and co-workers discovered a bacterial toxin that can base-edit mitochondrial DNA[37]. As for developing CRISPR-Cas systems for gene editing in mitochondria, excessive ROS as by-products of normal metabolism could be produced. Therefore, Acrs designed to function in a potentially oxidizing cellular environment should certainly avoid the redox effect in order to maximize their inhibitory activities. Herein, to perfect Acrs' behavior under various redox environments, we managed to generate a highly robust version of *Nme*AcrIIC1 with its redox-sensitive Cys residues replaced. The AcrIIC1 variant has a comparable binding capability to the HNH nuclease of Cas9 as wtAcrIIC1. Moreover, the inhibitory activity of the AcrIIC1 variant is much higher than wtAcrIIC1 under non-reducing environments, well suited to applications in cellular compartments with abnormally high ROS levels.

Additionally, we could take advantage of the redox regulation of Acrs to carry out tighter on-off control than direct regulation of the Cas

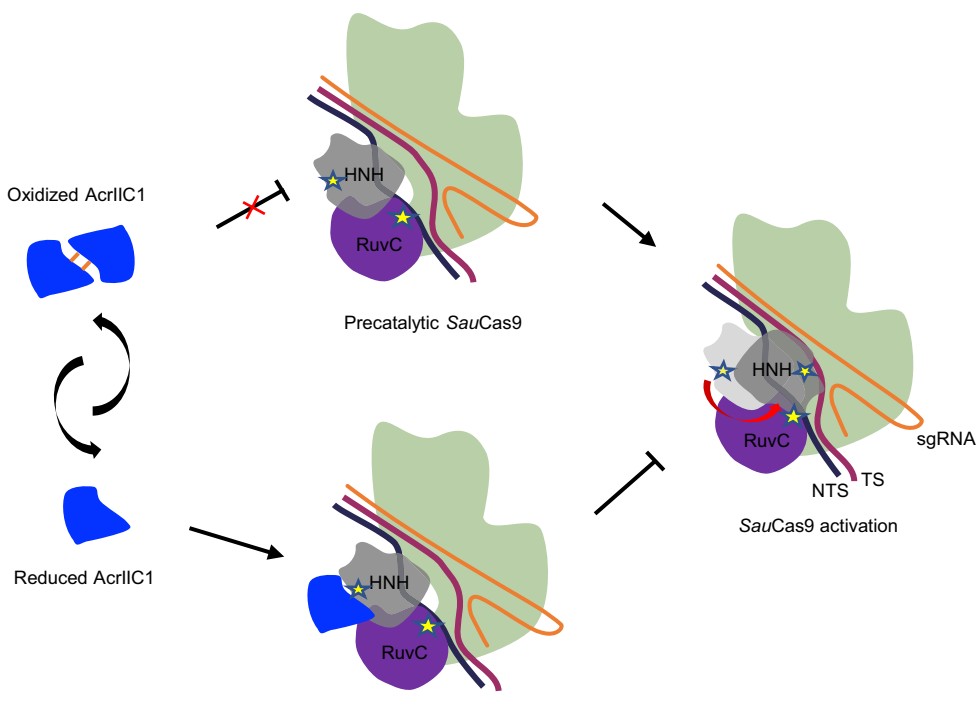

**Fig. 7 | Schematic model demonstrating how different redox states of AcrIIC1 affects the dynamics of Cas9 subunits and Cas9's activity.** Reduced AcrIIC1 would restrain the Cas9 complex at a precatalytic conformation through its interaction with the HNH domain and deactivate Cas9, while oxidized AcrIIC1 dimer would bury the contact interface with HNH, therefore loses its anti-Cas9 activity.

enzyme. Regulating Acrs to achieve more rapid and dynamic control of CRISPR-Cas activity has shown its feasibility[6]. Several methods for regulating Acr expression and activation have been developed, including transcriptional, post-transcriptional, optogenetic, and ligand-based strategies[16,38–40]. Based on our findings, we might offer another option for fine-tuning the CRISPR-Cas system by indirectly regulating the redox potential, which could potentially achieve an elaborated control of Cas9's cleavage activity. Utilizing the *Sau*Cas9 complex structure with its HNH domain bound to AcrIIA14[41], we propose a schematic model demonstrating how different redox states of AcrIIC1 would affect the dynamics of the Cas9 subunits and regulate Cas9's activity (Fig. 7). When sensing the environmental redox potential elevation, AcrIIC1 molecule can dimerize with its closest partner and then loses its contact interface with HNH, so HNH may complete its conformational rearrangement positioning its catalytic site onto the DNA substrate. In contrast, in the reduced cellular environment, the monomeric AcrIIC1 can bind to HNH and restrain the Cas9 complex at a precatalytic conformation without activity. Overall, our discovery of the naturally occurring redox switch for anti-CRISPR protein expands the regulatory toolset of the CRISPR system, and paves the road for further direction of engineering Acrs.

## Methods
### Protein expression and purification
The genes for *Nme*AcrIIC1, *Boe*AcrIIC1, *Pli*AcrIIC1, and *Nme*HNH were synthesized and sub-cloned into a modified pET-28a vector with a Protein G B1 domain (GB1) fused to the N-terminus, using cloning kit based on homologous recombination technology (Vazyme). The recombinant plasmids were transformed into *E. coli* BL21 (DE3) cells. Uniformed $^{15}N$- and $^{15}N$-, $^{13}C$- labeled proteins for NMR were obtained by growing the transformed cells at 25 °C in M9 minimal medium containing 0.5 g/L $^{15}NH_4Cl$, 2 g/L $^{13}C$-glucose supplemented with trace elements. Protein expression was induced with 0.5 mM IPTG when $OD_{600}$ of cells reached 0.6. After overnight, cells were harvested by centrifugation and resuspended in lysis buffer (20 mM Tris, 500 mM NaCl, 10 mM imidazole, pH 8.0). The cell lysate was extracted by sonication and cleared by centrifugation. The supernatant was applied onto a HisTrap column (GE Healthcare) equilibrated in lysis buffer and the desired protein was eluted at ~250 mM imidazole. The pooled protein was incubated with in-house purified Tobacco Etch Virus (TEV) protease overnight and loaded onto HisTrap again to remove the cleaved His-GB1 tag and dialyzed into NMR buffer (20 mM Tris, 100 mM NaCl, pH 7.4). Protein samples were further loaded onto a Superdex 75 Hiload 26/60 (GE Healthcare) equilibrated with NMR buffer. The reduced AcrIIC1s were supplemented with 1 mM DTT to maintain the reducing conditions. Purified proteins were then concentrated and stored at −80 °C for future use.

### NMR spectroscopy
NMR spectra were collected at 25 °C on Bruker Avance 600 MHz spectrometer equipped with HCN cryogenic probes and pulse-field gradients. NMR data were processed with NMRPipe[42] and analyzed with CCPNMR[43]. 1D $^1H$ spectra were collected to compare the difference between the monomer and dimer of AcrIIC1 orthologs. 2D $^1H$–$^{15}N$/$^{13}C$ HSQCs and 3D HNCO, HNCA, HNCACB, CBCA(CO)NH, HBHA(CO)NH, HCCH-TOCSY, and (H)CCH-TOCSY were performed to obtain the chemical shift assignments of backbone and side chain atoms[44]. 3D $^{13}C$-edited aliphatic and aromatic NOESY-HSQC (mixing time 100 ms) and $^{15}N$-edited NOESY-HSQC experiments (mixing time 100 ms) were collected to generate distance restraints for structural calculations.

### Phylogenetic analysis
The accession numbers of AcrIIC1 orthologs were acquired from a previous publication[8], and the protein sequence for each AcrIIC1 was downloaded from NCBI (https://www.ncbi.nlm.nih.gov/protein/). Proteins were aligned using ClustalOmega (https://www.ebi.ac.uk/Tools/msa/clustalo/) and an unscaled phylogenetic tree was built using MEGA software[45].

## Size-exclusion binding assays

About 100 µM *Nme*AcrIIC1 and 200 µM *Nme*HNH were combined to a volume of 400 µL, incubated for 10 min, and injected onto a Superdex 75 Increase 10/300 GL (Cytiva) with a 500 µL sample loop at 0.5 mL/min flow rate using an AKTA Purifier 10 FPLC system (GE Healthcare). The binding buffer is the same for NMR measurements, with 1 mM DTT supplemented only for the reduced *Nme*AcrIIC1. Fractions were analyzed on a 12% SDS-PAGE gel and stained with Coomassie blue.

## Ellman's assays

Ellman's reagent, DTNB (5,5′-dithio-bis-(2-nitrobenzoic acid)) (Bio-topped Life Sciences), was used to measure the free sulfhydryl groups in the protein solution. DTNB was dissolved to 100 mM in a reaction buffer of 20 mM Tris, pH 7.4, and 100 mM NaCl as a stock solution. To prepare the standards, DTT was dissolved in the reaction buffer with a gradually increased concentration (10, 20, 30, 40, and 50 µM). A final concentration of 10 mM DTNB was added and mixed with the standards. Absorbance at 412 nm (1 cm light path) was read by NanoDrop 2000C (Thermo Scientific) after 15 min of incubation. Each group was measured in double. Experiments were performed in triplicate. GraphPad Prism 8 was used to plot the absorbance values obtained for each standard against DTT concentrations, as shown in Supplementary Fig. 2b. The free sulfhydryl groups of the protein sample (concentration of ≤50 µM) was determined from the curve.

## Bio-layer interferometry (BLI) binding assays

The binding kinetics of the *Nme*HNH to different forms of *Nme*AcrIIC1 and variants were measured by BLI on a ForteBio Octet RED96 system (Pall ForteBio LLC). Assays were performed at 37 °C in a 96-well black flat bottom plate (Greiner Bio-One) with orbital shaking at 1000 rpm in an assay buffer of 20 mM Tris, 100 mM NaCl, pH 7.4 (with 1 mM DTT for the reduced AcrIIC1). Biotinylated *Nme*HNH was immobilized onto streptavidin biosensors (ForteBio) and incubated with varying concentrations of analytes in the solution. The kinetic experiments comprised 7 steps: (i) baseline acquisition: streptavidin biosensors were incubated in the assay buffer for 2 min; (ii) an auto-inhibition step: the analytes were incubated with the biosensors for 2 min; (iii) second baseline acquisition (6 min); (iv) loading: the biotinylated *Nme*HNH (10 µg/mL) was loaded onto the surfaces of biosensors until the response value accumulates to 1.2 nm; (v) third baseline acquisition (5 min); (vi) association: incubating biosensors with different concentrations of analytes for 3 min to measure $K_{on}$; (vii) dissociation: incubating biosensors with the assay buffer for 5 min to measure $K_{off}$. The resulting curves were corrected by subtracting the blank reference, then fitted to a global fit algorithm using a 1:1 binding assumption to determine the $K_d$ ($K_{off}/K_{on}$) by the ForteBio Data Analysis software 9.0.

## In vitro transcription and purification of sgRNA

In vitro transcription was carried out with in-house purified T7 RNA polymerase using a standard protocol. The template (dsDNA) for sgRNA was generated by PCR. The sequence of the sgRNA is 5′-GG GUGCGCGGCGCAUUACCUUUACGUUGUAGCUCCCUUUCUCAUUUC GGAAACGAAAUGAGAACCGUUGCUACAAUAAGGCCGUCUGAAAAGA UGUGCCGCAACGCUCUGCCCCUUAAAGCUU-3′. Transcription reactions were performed at 37 °C for 4 h in buffer containing 0.1 M HEPES-K, 12 mM MgCl₂, 30 mM DTT, 2 mM spermidine, 2 mM each rNTP (rATP, rUTP, rGTP, rCTP), 100 µg/mL T7 polymerase, and 500 nM transcription template. The sgRNA was purified using an 8% denatured polyacrylamide gel and then ran through the Elutrap system. Finally, the sgRNA was resuspended in diethylpyrocarbonate (DEPC) H₂O, stored at −80 °C.

## In vitro DNA cleavage assays

For 10 µL of the reaction system, 2 µM Cas9 RNP complex (2 µM Cas9 mixed with 2.5 µM sgRNA) was pre-formed in the reaction buffer (20 mM HEPES, pH 7.5, 100 mM KCl, 10 mM MgCl₂, and 5% (w/v) glycerol with or without 5 mM DTT for reducing condition and oxidizing condition, respectively). About 0–4 µM AcrIIC1 was added into the system with the molar ratios of Cas9 to AcrIIC1 ranging from 1:0 to 1:2. The target DNA containing a protospacer sequence with PAM motif (5′-TGCGCGGCGCATTACCTTTACGCCGGATTGCTG-3′) was cloned and amplified using PCR. About 400 ng target DNA was added last and the cleavage reaction was performed at 37 °C for 30 min. The cleavage products were treated with 50 µg/mL proteinase K (Solarbio Life Sciences) at 56 °C for 30 min. Finally, the reaction products were separated and visualized using 1% agarose gels stained with ethidium bromide. For assays shown in Fig. 5d, to mimic the oxidizing condition when experiencing oxidative stress, a fivefold molar ratio of H₂O₂ relative to AcrIIC1 variants was added. The reaction was performed for 50 min to compensate for the decreased activity of Cas9 potentially caused by H₂O₂ damage.

## Structure determination by NMR

We used solution NMR for the structural calculation of the reduced *Nme*AcrIIC1. 100 starting structures of the reduced *Nme*AcrIIC1 were initially calculated using CYANA[46,47] and then refined with AMBER[48]. Only distance restraints obtained from 3D ¹³C/¹⁵N-edited NOESY-HSQC experiments were used as initial inputs for the first several rounds of calculation in CYANA. In the following iterations, 92 torsion angle restraints for both φ and ψ predicted by TALOS+ were included[49], based on the secondary structure identified from the early structure ensembles. Hydrogen-deuterium exchange experiments were carried out to identify the protected amides and confirm the hydrogen bond network in secondary structures. The chemical shifts for $C_\alpha$ and $C_\beta$ of C17 and C80 clearly show their reduced states, so we kept them as cysteines along with the structural calculation. The 100 structures with the lowest target functions were refined in AMBER 14.0 using a 20-ps restrained simulated annealing protocol. The annealing was done with the ff14SB force field and generalized Born solvation model, restrained by the same distances and torsion angles used in CYANA with force constants of 35 and 20 kcal mol⁻¹ Å⁻¹, respectively. In addition, chirality restraints were applied along the main chain to prevent inversions of chirality at high temperatures. The 20 conformers with the lowest AMBER energy were finally selected.

## ¹⁵N backbone relaxation measurements

The backbone ¹⁵N relaxation parameters longitudinal relaxation rates $R_1$ [$1/T_1$], transverse relaxation rates $R_2$ [$1/T_2$], and steady-state heteronuclear NOEs were measured using standard pulse sequences[50]. Experiments were performed on Bruker Avance 600 MHz NMR spectrometer at 25 °C. Spectral widths of 8417.5 and 1702.9 Hz were used for ¹H and ¹⁵N dimensions, respectively. For the $T_1$ and $T_2$ measurements, 1024 (¹H) and 128 (¹⁵N) complex data points were collected with 16 transients and a recycle delay of 3 s. For the reduced *Nme*AcrIIC1, the delays used for the $T_1$ experiments were 10 (×2), 60, 120, 200, 300, 500, 800, 1100, 1500, and 2000 ms, and those used for the $T_2$ experiments were 10 (×2), 30, 50, 80, 120, 160, 200, 250, and 300 ms. For the oxidized *Nme*AcrIIC1, the delays used for the $T_1$ experiments were 10 (×2), 60, 120, 200, 300, 500, 800, 1100, 1500, 2000, and 2500 ms, and those used for the $T_2$ experiments were 10 (×2), 30, 50, 80, 120, 160, 200, and 250 ms. The relaxation rate constants were obtained by fitting the peak intensities to a single exponential function using the nonlinear least squares method. For the {¹H}-¹⁵N Het-NOE measurements, two spectra were performed in the presence and

absence of a 3 s proton pre-saturation period prior to the $^{15}$N excitation pulse and using recycle delays of 2 and 5 s, respectively. 32 transients were used for each experiment. Het-NOE values were calculated from the ratio of peak intensities with and without proton saturation. The rotational correlation times values ($\tau_c$) were estimated by the following equation:

$$\tau_C \approx \frac{1}{4\pi\nu_N}\sqrt{6\frac{R_2}{R_1}-7} \tag{1}$$

where $\nu_N$ is the $^{15}$N resonance frequency (in Hz)[51].

### Crystal growth, X-ray diffraction, and structural determination
The oxidized *Nme*AcrIIC1 was concentrated to 20 mg/mL in buffer 20 mM Tris, pH 7.4, 100 mM NaCl for crystallization. 1 μL protein and 1 μL of reservoir buffer (1.8 M $(NH_4)_2SO_4$, 0.1 M Bis-Tris at pH 6.5 and 2% (v/v) polyethylene glycol monomethyl ether 550) were mixed for crystallization at 16 °C in a sitting drop manner. Crystals appear in about one week. Single crystals were immersed in mother liquor containing 20% (v/v) glycerol and flash-frozen in liquid nitrogen. Diffraction data sets were collected at the beamline BL18U1 of the SSRF (Shanghai Synchrotron Radiation Facility)[52]. Data sets were processed with XDS and autoPROC software suite[53,54]. The structure from PDB entry 5VGB was used as a search template to solve the structure of the oxidized *Nme*AcrIIC1 by molecular replacement in the Phenix suite[55]. All structure models were manually adjusted in Coot and refined with Phenix refine program[55,56]. All the figures of protein structure were prepared with the software PyMol (http://www.pymol.org).

### pKa determination by NMR
$^1$H-$^{13}$C HSQCs were used to acquire the chemical shifts of $C_\beta$ for Cys17 and Cys80 during pH titration. NMR sample with gradually increased pH values (6.0, 6.5, 7.0, 7.5, 8.0, 8.5, 9.0, and 9.5) was obtained by buffer exchanging into 50 mM Tris, 100 mM NaCl adjusted at corresponding pH. Chemical shifts were calibrated using DSS (2,2-Dimethyl-2-Silapentane-5-Sulfonic acid), and plotted against the pH values. GraphPad Prism 8 was used to analyze the pH dependence of chemical shifts, the resulting sigmoidal curve was subjected to nonlinear least squares fitting to the modified Henderson-Hasselbalch equation:

$$\delta = \delta_{low} - \frac{\delta_{low} - \delta_{high}}{1 + 10^{n(pK_a - pH)}} \tag{2}$$

where the pKa is the ionization constant, $\delta_{low}$ is the low pH chemical shift plateau, $\delta_{high}$ is the high pH chemical shift plateau, and n is the apparent Hill coefficient[57].

### Antimicrobial susceptibility assays
First, using the two-plasmid CRISPR system pCas-pSG reported by Dr. Quanjiang Ji's group[23], we sought to delete the kanamycin resistance gene *kmR* carried with the pSG plasmid in the industrial *E. coli* strain BL21 and verified the genome-editing ability by observing a larger inhibition zone of the bacterial lawn of growth, compared to the negative control carrying no spacer gene. To do this, the pCas plasmid was first transformed into the *E. coli* strain to obtain the pCas-harboring strain, and the cells containing the pCas plasmid were collected and prepared as competent cells. Then, the *kmR* spacer-introduced plasmid pSG-KMR was transformed into the aforementioned pCas-harboring competent cells, and the freshly cultured *E. coli* cells were suspended, diluted, and plated with a kanamycin (100 μg) tablet placed in the center at 30 °C overnight. The diameter of the inhibition ring was measured after 20 h incubation. A control experiment using bacterial cells harboring pSG plasmid without a spacer gene was carried out at the same time. For experimental details, we followed Ji group's publication[23]. To make a quantitative comparison

of the size of the inhibition zone, we controlled the amount of the plasmids transformed, the cell turbidity and the culture volume coated, and the incubation time to be precisely consistent among different groups of experiments. Replication of each group of experiments more than three times yields consistent data.

Second, to assess the anti-CRISPR activity of AcrIIC1, the plasmid expressing WT or mutant AcrIIC1 was transformed into the pCas-harboring competent cells and prepared as a new kind of competent cell harboring both pCas and pAcrIIC1 plasmids. Then, the plasmid pSG-KMR or pSG was transformed into the aforementioned pCas-pAcrIIC1-harboring competent cells and the *E. coli* cells were cultured. The expression of AcrIIC1 was induced with 0.5 mM of IPTG when the cell density reached an $OD_{600}$ of 0.2. Then as above, when $OD_{600}$ reached 0.6, the cells were suspended, diluted, and plated. For assays under oxidizing environments, 1 mM $H_2O_2$ was added along with IPTG to induce the expression of the oxidized AcrIIC1 in the cell.

To confirm the overexpression of AcrIIC1 and its redox state, 1 mL of culture after IPTG induction for 2–3 h was centrifuged, and then the cells were resuspended in 100 μL loading buffer. The AcrIIC1 expression and its redox state were analyzed on a 12% native PAGE gel.

### Reporting summary
Further information on research design is available in the Nature Portfolio Reporting Summary linked to this article.

## Data availability
All data generated or analyzed in this study are included in the main text or the supplementary materials. Atomic coordinates and NMR data for the reduced *Nme*AcrIIC1 have been deposited in the Protein Data Bank (PDB) under entry ID 7X31 and Biological Magnetic Resonance Bank (BMRB) under entry ID 36471. Atomic coordinates and structure factors for the crystal structures of the oxidized state of *Nme*AcrIIC1 have been deposited in the PDB under entry ID 7X4B. One other published PDB code cited in this paper is 5VGB. The coordinates for the structural models generated in this study are provided as supporting information. The source data are provided as a Source Date file with this paper. Source data are provided with this paper.

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

## Acknowledgements

We thank Dr. Chunfu Xu for his help on the structural model using AlphaFold and Dr. Quanjiang Ji for generously providing the pCas-pSG plasmids and helpful discussions. We thank Dr. Peter Hsu for his critical reading of the manuscript. We also thank the NMR facility of the National

Center for Protein Sciences at Peking University for assistance with NMR and Dr. Hongwei Li for help with data collection, and the staff members of the National Facility for Protein Science in Shanghai, Zhangjiang Laboratory for providing technical support and assistance in NMR and BLI data collection. This work was supported by the National Natural Science Foundation of China (31900863 to F.Y., joint funds U1932204 to S.Z., and 31800630 to Y.Z.), the Natural Science Foundation of Heilongjiang Province of China (LH2021C049 to F.Y.), the Fundamental Research Funds for the Central Universities (HIT.NSRIF202210 to F.Y.), CAS project for Young Scientists in Basic Research (YSBR-009 to S.Z.), and the funds from the International Cooperation Division at Harbin Institute of Technology.

## Author contributions

F.Y. conceived the project, performed the NMR experiments and analysis, and wrote the paper. Y. Zhao, J.H., and S.-S.Y. contributed to the sample preparation, NMR analysis, and in vitro assays. J. Zhong and S.W. performed the DNA cleavage assays. J.L. solved the crystal structure. Y.J., F.J., B.R., Y. Zhu, J. Zhang, and Y.X. helped in the protein purification and binding analysis. R.Z. performed the phylogenetic analysis. J. Zhong, H.C., and F.H. contributed to the antimicrobial susceptibility assays. Z.H., S.Z., and F.Y. supervised the project.

## Competing interests

The authors declare no competing interests.
