## [Peer Review File · Nature Communications]

Reviewer comments, first round review:

Reviewer #1 (Remarks to the Author):

In this paper, Zhao et al discovered new redox switch for AcrIIC1 that inhibits the Cas9 activity via redox-mediated stoichiometric interconversion. They solved dimeric NmeAcrIIC1 structure that was formed by two disulfide bonds between Cys17 and Cys80. They also confirmed the regulation of the inhibitory activity of AcrIIC1 by interchange of monomer-dimer formation using mutagenesis studies. Finally, they insisted that this might be novel redox-dependent regulation mechanism of Acr.

Although this paper contains interesting features finding tentative noble mechanism of activity regulation of Acr by redox environment, complex structure of AcrIIC1 and HNH domain of Cas9 is not novel and the study has some limitations.

1. Oxidized dimeric structure of AcrIIC1 might be artifact form produced in high concentration of AcrIIC1 protein. The electron density map at disulfide bonds provided at supplementary Figure 4 is not clear although the structure is very high resolution. Even sigma cut-off value is not provided.
2. Structural information of AcrIIC1 and HNH domain of Cas9 was already provided by Doudna group at 2017 (Cell 7:170). Oxidized form is easily detected in the natural environment. The reason why previous study did not find oxidized form of AcrIIC1 have to be analyzed and discussed.
3. Cysteine residues involving in the formation of disulfide bonds are not conserved in most of the species. These cysteines are conserved at several specific species. Therefore, this regulation mechanism might not be applied for all the AcrIIC1 family. The highlighting position for conserved residues is wrong at Supplementary Figure 1.
4. The biological implication of this redox switch for regulation of AcrIIC1, which might be interesting, is not studied in this moment and should be proved for publishing this high impact journal.

Reviewer #2 (Remarks to the Author):

The authors present a structural and biochemical characterization of the redox-dependent interaction of an anti-CRISPR (Acr) protein with NmeCas9. They show, for the first time, that the inhibitory action of AcrIIC1 is redox-controlled due to a change in oligomeric state that occludes the binding interface with NmeHNH. Using NMR, X-ray crystallography, and BLI, the authors present the structural and binding signatures of the redox-dependent Acr-HNH complex. They extend this work to biochemical assays that corroborate the structural findings, namely that AcrIIC1 is unable to inhibit NmeCas9 under oxidizing conditions.

In general, I find the premise of the work to be strong and the data to be of high quality and rigorously collected. I especially appreciate the internal consistency of several techniques ranging from basic biochemistry to high-level structural biology. This report is novel and has potential to usher in further studies of redox-dependent conformational switching in CRISPR biology, informing us of the role of redox in selective control of Cas9s. There remain several points that should be addressed before the paper can be published.

- 1) The majority of the results presented by the authors describe the "air oxidized" state of AcrIIC1, rather than a sample treated with oxidant. In several cases, there is clear evidence for only partial oxidation of the critical Cys residues (see Figure 4d, Supplementary Figure 1, Supplementary Figure 4). It is also known from other studies, some of which include mass spectrometry data, that Cys residues may be only partially oxidized under benchtop conditions.

While the authors acknowledge this, provide convincing evidence for the redox-driven switch regardless, and conduct some experiments in the presence of peroxide, they should comment (or contrast) the meaning of, and any discrepancies between, the two conditions a bit more.

2) On a somewhat related note, there is no mention of redox potentials accessed during the course of these experiments, or how these conditions would compare to those found in a cell. Some swings in biological redox potentials are quantified in the literature, as is their effect on protein stability and function. If there is any context for the experimental conditions reported here that can be discussed based on this knowledge, it would be a welcome addition.

3) Page 7, lines 124-128 – the authors discuss the lack of inhibition by air oxidized NmeAcrIIC1 toward NmeCas9. Has this effect been observed in the literature for any other Cas9 species? It seems reasonable to assume that if this manuscript is the first report of redox-switching governing the efficacy of Acr proteins, then prior biochemical assays of Cas9 inhibition by Acrs may not have controlled for redox state. Does this mean that some prior assays could be clouded by the uncertainty in the redox conditions? If not, why not? There are moments when the authors discuss how wide-ranging such a mechanism might be for Acr proteins, but this seems like an important point of context here.

4) It is intriguing that both the monomeric (reduced) and dimeric (oxidized) forms of AcrIIC1 are rigid structures (via NMR relaxation) when it is widely accepted that Cas9 proteins themselves are highly dynamic. It is possible the motion exists on other timescales, but it might be worth mentioning how/if this Acr is proposed to hinder dynamics of the system when interacting.

5) Page 10, line 219 – the term “design” here seems like a bit of an overstatement, since the two Cys residues are native to the Acr protein and this study is simply removing them. It is very strong proof that these residues are involved in the redox switching, but in my opinion, “design” implies more than basic mutagenesis. Perhaps if Cys residues were inserted at another point within the protein with some measure of the same functionality, it would apply.

6) Figure 1, panel a – the authors should show the two chromatograms overlaid on top of each other to better demonstrate the peak shift. Were any experiments with molecular weight standards carried out by SEC?

7) Figure 2 – panels A and B seem like they should be swapped.

8) Figure 2 – on a related note in panel b, are the traces concentration normalized? This information might be useful in the caption to more easily interpret the shifts (or lack thereof) in the red traces corresponding to the complexes.

9) Figure 4, panel b – this spectrum should be compared to an oxidized spectrum, similarly to the comparison in panel a.

10) Figure 5, panel b – it is not clear from the caption which conditions correspond to the BLI plot. In the “SS” mutant, the authors note binding under both redox conditions, due to the removal of Cys and prevention of the dimer. However, only one set of traces appears and the redox state is not indicated. Do data exist for both redox states?

11) Supplementary Figure 4 – it is hard to imagine such a small error (± 0.054) in the pKa determination of Cys17 when the pH plot has not yet begun to level off in the high pH regime.

12) There are many grammatical and typographic errors throughout the manuscript. Recommend carefully reading through again to correct these issues.

Point-by-point responses to the reviewers' comments:

* The reviewers' comments are in *Italic*. Author's responses are in blue. All revisions are highlighted in red in the revised manuscript.

Reviewer #1 (Remarks to the Author):

In this paper, Zhao et al discovered new redox switch for AcrIIIC1 that inhibits the Cas9 activity via redox-mediated stoichiometric interconversion. They solved dimeric NmeAcrIIIC1 structure that was formed by two disulfides bonds between Cys17 and Cys80. They also confirmed the regulation of the inhibitory activity of AcrIIIC1 by interchange of monomer-dimer formation using mutagenesis studies. Finally, they insisted that this might be novel redox-dependent regulation mechanism of Acr. Although this paper contains interesting features finding tentative noble mechanism of activity regulation of Acr by redox environment, complex structure of AcrIIIC1 and HNH domain of Cas9 is not novel and the study has some limitations.

REPLY: We sincerely thank the reviewer for his/her insightful suggestions provided below, which are highly important for us to improve our manuscript. Accordingly, substantial experiments were performed to address the reviewer's concerns as detailed below.

1. Oxidized dimeric structure of AcrIIIC1 might be artifact form produced in high concentration of AcrIIIC1 protein. The electron density map at disulfide bonds provided at supplementary Figure 4 is not clear although the structure is very high resolution. Even sigma cut-off value is not provided.

REPLY: We thank the reviewer for raising this important point. We totally agree with the reviewer that if the oxidized dimer was only available under crystallography condition where almost 2 mM of AcrIIIC1 molecule is used for crystal growth, the artifact might be a concern. Importantly, in addition to the dimeric structure identified in the crystal, we did observe that the oxidized dimer of AcrIIIC1 can be formed under several other experimental conditions where nM- μ M of protein was used. We list the concentrations of protein and the methods for monitoring the dimer formation in different assays below.

(1). Oxidized dimeric AcrIIIC1 can be converted from the reduced AcrIIIC1 with a protein concentration of **20-100 μ M**, based on SEC results and 2D ^1H - ^{15}N HSQC spectra, as shown in **Fig. 1a**. Of note, the chemical shifts for C_β of Cys17 and Cys80 of the oxidized AcrIIIC1 fall into a typical range for oxidized cystine (36.3 ppm ~ 47.9 ppm) instead of reduced cysteine (24.9 ppm ~ 29.7 ppm)¹, shown as the black peak strips in **Fig. S4b**. Together with the ^{15}N relaxation data (**Fig. 3e**), solution structure supports dimeric AcrIIIC1 as seen in the crystal.

(2). **50 μ M** of oxidized dimeric AcrIIIC1 can be observed from native PAGE gel as well (**Fig. S2a**).

(3). As for the biochemical evidence supporting oxidized dimeric AcrIIIC1 formation at a low concentration of protein, less than **4 μM** of AcrIIIC1 can be oxidized to its inactive form from DNA cleavage assay, shown as **Fig. 2d,4d,5d**; **31-250 nM** of AcrIIIC1 can be oxidized into a form losing binding capability with Cas9, based on BLI assay results, shown as **Fig. 2e**.

(4). Our newly collected in cell data shows that under environmental H_2O_2 stimulus, AcrIIIC1 can be oxidized into its inactive form *in vivo* (**Fig. S7**), with a protein concentration on an nM-uM scale.

Collectively, these data support that the oxidized dimeric form of AcrIIIC1 is not an artifact formed at a high concentration of protein, but a natural conformational state robustly formed under oxidative conditions at low concentration, both *in vitro* and *in vivo*.

As for the electron density map in the original **Fig. S4** (**Fig. S5** in the revised manuscript), we sincerely apologize for the confusion, and thank the reviewer for bringing it up. To make it clear, we prepared a new figure with a sigma cut-off value noted in the figure legend. The electron density map is shown with a cut-off of 2 sigma (**Fig. R1-1A**). In addition, we examined the existence of disulfide bonds between Cys17 and Cys80 by the matching degree between the structural model and the electron density map. As we refined the structural model with reduced cysteines for Cys17 and Cys80 against the electron map, one pair of Cys17-Cys80 forms a disulfide bond automatically due to the constrained orientation by the map and the short distance between the two sulfur atoms (lower pair in **Fig. R1-1A**). While the other pair of Cys17-Cys80 (upper pair in **Fig. R1-1A**) can adopt either reduced or disulfide bonded conformation. Thus, we examined the Fo-Fc map (at 2 sigma cut-off) calculated with the reduced Cys17-Cys80 structural model for the upper pair. As shown in **Fig. R1-1B**, extra un-occupied density between the two free cysteines (the orange region in **Fig. R1-1B**) does appear, which can be well fitted by a disulfide bond between Cys17-Cys80 (**Fig. R1-1C**). This data supports that Cys17 and Cys80 form disulfide bonds in the crystal. However, the occupancy of both disulfide bonds is below 100%, supporting the dynamic redox switch of the protein.

Figure R1-1 A validation of the disulfide bonds between Cys17 and Cys80.

A, Revised map adapted from **Fig. S5a**. 2Fo-Fc electron density map is contoured at 2.0σ . The upper pair of Cys17-Cys80 can adopt two conformations, either free cysteine or disulfide bonded.

B, The Fo-Fc map (shown at 2 sigma level) calculated with the reduced Cys17-Cys80 structural model reveals un-occupied density around these two residues.

C, The extra density can be well fitted with a disulfide bond between Cys17 and Cys80.

2. Structural information of AcrIIIC1 and HNH domain of Cas9 was already provided by Doudna group at 2017 (Cell 7:170). Oxidized form is easily detected in the natural environment. The reason why previous study did not find oxidized form of AcrIIIC1 have to be analyzed and discussed.

REPLY: We thank the reviewer for raising this concern. To our knowledge, protein purification for structural biology study routinely adds excess DTT or TCEP to avoid potential mismatched intermolecular disulfide bond linkage, especially for proteins without prior knowledge of redox regulation involved. We think this might be one possible reason why previous studies did not find the oxidized AcrIIIC1. Additionally, we checked 9 publications that investigated AcrIIIC1 experimentally, and found that reducing agents were indeed used for their studies. For clarity, we list the reducing buffer conditions for protein purification or assays used in previous researches in **Supplementary Table 3**²⁻¹¹. Moreover, we assessed the redox potentials of our buffers and compared them with intracellular values reported previously, and discussed why *in vivo* assays didn't identify the oxidized form of AcrIIIC1 a little. We added the following discussion on page 14 in the revised manuscript to address the reviewer's concern:

*“One may wonder why previous studies haven't identified the oxidized form of NmeAcrIIIC1. After surveying 9 publications that investigated AcrIIIC1 experimentally, including structural studies, in vitro DNA cleavage or binding assays, we find that previous in vitro assays were all performed with reductants maintaining reducing experimental conditions (listed as **Supplementary Table 3**). For example, in the structural study of NmeAcrIIIC1 complexed with HNH by Doudna group in 2017, they kept 0.5 -1 mM TCEP-NaOH in all purification buffers and 1 mM DTT or 1mM TCEP-NaOH for all in vitro cleavage and binding assays. It could be the reducing experimental condition that precluded the chance of forming the oxidized AcrIIIC1. As for previous in vivo assays, since only apparent activity of AcrIIIC1 can be observed without characterization of the molecular state, it's hard to tell the exact redox state of AcrIIIC1 molecules in cell. To assess the intracellular state of AcrIIIC1, we measured the redox potentials for the reducing buffers we used (listed as **Supplementary Table 4**), and compared them with the intracellular redox potentials reported previously. It seems that normal cytoplasmic values (-260 mV to -200 mV) fall within the range of our reducing buffers (-360 mV to -200 mV), indicating most of the AcrIIIC1 molecules should be at their reduced state under normal physiological conditions. Under oxidative stress, local redox potential elevation in cell is not easy to measure at this point, due to the lack of a stable redox sensor working in this condition. However, it was reported that cells exposed to a more oxidized potential (0 mV) had stimulated H₂O₂ production, which can induce the inactive dimer of AcrIIIC1 based on our in vitro and in vivo data (**Supplementary Fig. 7**).”*

For the reviewer's convenience, we paste the newly added **Supplementary Table 3** here, as below:

Supplementary Table 3: Reducing buffer conditions used in previous AcrIIc1 studies *in vitro*.

Publications	Protein purification	In vitro DNA cleavage assay	Binding assay with Cas9
Mathony, J. et al. Computational design of anti-CRISPR proteins with improved inhibition potency. Nat. Chem. Biol. 16,725-730(2020).	—	20 mM Tris, pH 7.5, 100 mM KCl, 5 mM MgCl ₂ , 5% glycerol, 1 mM DTT	—
Kim, Y. et al. Anti-CRISPR AcrIIc3 discriminates between Cas9 orthologs via targeting the variable surface of the HNH nuclease domain. FEBS J. 286, 4661-4674(2019).	—	20 mM sodium phosphate, pH 6.5, 50 mM NaCl, 1 mM PMSF 2 mM BME	20 mM sodium phosphate, pH 7.4, 100 mM NaCl, 5 mM BME
Zhu, Y. et al. Diverse Mechanisms of CRISPR-Cas9 Inhibition by Type IIC Anti-CRISPR Proteins. Mol. Cell 74, 296-309.e7(2019).	20 mM Tris-HCl, pH 7.5, 300 mM NaCl, 2 mM MgCl ₂ , 1 mM DTT	20 mM Tris-HCl, pH 7.5, 100 mM KCl, 5 mM MgCl ₂ , 5% glycerol, 5 mM DTT	20 mM Tris-HCl, pH 7.5, 300 mM NaCl, 2 mM MgCl ₂ , 1 mM DTT
Phaneuf, CR. et al. Ultrasensitive Multi-Species Detection of CRISPR-Cas9 by a Portable Centrifugal Microfluidic Platform. Anal Methods. 11, 559-565(2019).	50 mM Tris-HCl, pH 7.5, 500 mM NaCl, 10% glycerol, 0.5 mM TCEP	—	—
Edraki, A. et al. A Compact, High-Accuracy Cas9 with a Dinucleotide PAM for In Vivo Genome Editing. Mol Cell. 73, 714-726.e4(2019).	50 mM Tris-HCl, pH 7.5, 500 mM NaCl, 5 mM imidazole, 1 mM DTT	—	—
Seamon, KJ. et al. Versatile High-Throughput Fluorescence Assay for Monitoring Cas9 Activity. Anal Chem. 90, 6913-6921(2018).	50 mM Tris-HCl, pH 7.5, 100 mM NaCl, 0.5 mM TCEP	—	—
Rousseau, BA. et al. Programmable RNA Cleavage and Recognition by a Natural CRISPR-Cas9 System from Neisseria meningitidis. Mol Cell. 69,906-914.e4(2018).	20 mM HEPES, pH 7.5, 300 mM NaCl, 0.5 mM DTT	20 mM HEPES, pH 7.5, 50 mM KCl, 0.1 mM EDTA, 10 mM MgCl ₂ , 0.5 mM DTT	—
Harrington, L. B. et al. A Broad-Spectrum Inhibitor of CRISPR-Cas9. Cell 170, 1224-1233.e15 (2017).	50 mM Tris-HCl, pH 7.5, 20 mM imidazole, 500 mM NaCl, 1 mM PMSF, 0.5 mM TCEP-NaOH	20 mM Tris-HCl, pH 7.5, 100 mM KCl, 5 mM MgCl ₂ , 5% glycerol, 1 mM DTT	20 mM HEPES-NaOH, pH 7.5, 150 mM NaCl, 1 mM TCEP-NaOH
Pawluk, A. et al. Naturally Occurring Off-Switches for CRISPR-Cas9. Cell 167, 1829-1838.e9(2016).	10 mM Tris, pH 7.5, 250mM NaCl, 5mM BME	20 mM HEPES-KOH, pH 7.5, 150 mM KCl, 10% glycerol, 10 mM MgCl ₂ , 1 mM DTT	—

3. Cysteine residues involving in the formation of disulfide bonds are not conserved in most of the species. These cysteines are conserved at several specific species. Therefore, this regulation mechanism might not be applied for all the AcrIIIC1 family. The highlighting position for conserved residues is wrong at Supplementary Figure 1.

REPLY: This is a good point, very important to improve our manuscript to be more precise. Indeed, out of 11 identified AcrIIIC1 homologs from different species, only 4 AcrIIIC1 possess the conserved Cys17 and 3 AcrIIIC1 possess the conserved Cys80. Although the other two AcrIIIC1 orthologs (from *Boe* and *Pli*) containing two conserved Cys residues were proved by us to undergo similar redox regulation as *NmeAcrIIIC1*, we totally agree with the reviewer that this regulation mechanism might not be applied for all the AcrIIIC1 family. Accordingly, we revised the first paragraph in the section of Discussion as follows:

“It’s very possible that the redox switch identified from AcrIIIC1 may serve as a common regulation mechanism shared by other Acr proteins. It’s been only less than ten years since the first anti-CRISPR protein was identified, there is so much to be uncovered in this field, including more members in AcrIIIC1 family and other Acr proteins that might possess redox-sensitive Cys residues.”

Regarding the mis-highlighting in **Fig. S1**, we sincerely apologize and we’ve corrected it in the revised **Fig. S1**.

4. The biological implication of this redox switch for regulation of AcrIIIC1, which might be interesting, is not studied in this moment and should be proved for publishing this high impact journal.

REPLY: We sincerely thank the reviewer for providing this suggestion. We were thinking of some ideas to test the biological implication of our findings, but somehow all the tests were stalled by different sorts of difficulties. The reviewer’s suggestion gave us the push and we are glad we do get some promising biological data at this stage. As the reviewer points out, this would increase the impact of our work. *In vivo* experiments in the original organism *Neisseria meningitidis* were not an option for us, because it would need a high-level biosafety laboratory that is not available around. Alternatively, we managed to perform antimicrobial susceptibility assays to access the redox regulation of AcrIIIC1 in *E. coli*, which is also a gram-negative bacterium resembling *Nme* in several ways. Briefly, we established an antimicrobial susceptibility assay to assess the anti-CRISPR activity of wtAcrIIIC1 and AcrIIIC1-SS in cell by observing the size of the inhibition zone of the bacterial growth. Our results demonstrate that, in cell, the inhibitory activity of wtAcrIIIC1 varies in response to environmental redox change. However, AcrIIIC1-SS (a mutant with redox switch abolished) remains constantly high activity resistant to environmental redox change. We’ve appended the related results to the section “AcrIIIC1 variant with a robust anti-Cas9 activity in diverse redox environments” in Results, as follows:

“To further investigate the redox regulation of wtAcrIIIC1 biologically and to examine how robust AcrIIIC1-SS behaves in various redox environments in cell, we established an antimicrobial

susceptibility assay to qualitatively assess the anti-CRISPR activity of AcrIIIC1 in *E. coli*. Taking advantage of the pCas-pSG-mediated constitutive expression of Cas9 and an sgRNA targeting the kanamycin resistance gene, we succeeded to observe a larger inhibition ring around the paper disk impregnated with kanamycin on the bacterial lawn of growth relative to the negative control, as shown in **Supplementary Fig. 7a,c** panel 1,2. Very importantly, co-expression of wtAcrIIIC1 induced by IPTG restored the inhibition zone to a comparable size as where no CRISPR-Cas9 functions (**Supplementary Fig. 7a,c** panel 1,3,4), strongly indicating that Cas9 is mostly inhibited by the over-expressed wtAcrIIIC1. By contrast, the inhibition zone expanded to a similar size as where wtAcrIIIC1 was not expressed, when the bacterial cells were exposed to 1 mM H₂O₂ during wtAcrIIIC1 expression, implying substantial activity loss of wtAcrIIIC1 under the oxidative environment (**Supplementary Fig. 7a,c** panel 2,6). Moreover, native PAGE gel confirmed almost complete conversion of wtAcrIIIC1 to its oxidized dimeric form in cell under H₂O₂ stimulus (**Supplementary Fig. 7d**). Not surprisingly, when co-expressed with AcrIIIC1-SS, the inhibition zone remained the same size with or without H₂O₂ stimulus, as shown in **Supplementary Fig. 7b,c** panel 3,4,5,6, stating that AcrIIIC1-SS behaves as a superior Cas9 inhibitor with more stable activity *in vivo*.”

For the reviewer’s convenience, we adapted **Supplementary Fig. 7** as **Fig. R1-4**, as below:

C

D

E

Figure R1-4. Antimicrobial susceptibility assays assessing wtAcrIIIC1 and AcrIIIC1-SS activities *in vivo*.

A, Panel 1,2, assays without wtAcrIIIC1 expression. The gene-editing ability of the two-plasmid system pCas-pSG was confirmed by a significant increase of the size of the inhibition ring relative to a negative control, with a representative diameter of 21 mm vs 17 mm. Panel 3,4, assays with wtAcrIIIC1 induced by IPTG. Co-expression of wtAcrIIIC1 restored the inhibition ring because it binds to and deactivates Cas9. Panel 5,6, assays with wtAcrIIIC1 induced and 1 mM H₂O₂ stimulus. AcrIIIC1 failed to restore the inhibition ring under H₂O₂ stimulus due to the loss of activity under the oxidative environment.

B, Co-expression of AcrIIIC1-SS can restore the inhibition ring under various redox environments because its activity is resistant to redox fluctuation. Assay panels are in the same order as **A**.

C, Statistical analysis of the inhibition zone diameters from **A,B**. The left part shows the size of the inhibition ring varies in response to environmental redox change when wtAcrIIIC1 is co-expressed with Cas9. The right panel shows the size of the inhibition ring remains constant resistant to environmental redox changes when AcrIIIC1-SS is co-expressed with Cas9. Lines in the plots indicate means, dots indicate individual data points from three independent experiments, and every dot is calculated by averaging three rings' diameters from one plate. Error bars represent SD. **P < 0.01, ***P < 0.001 by unpaired t-test.

D, Native PAGE results show the overexpression and oligomeric state of wtAcrIIIC1 in cell (Lane 2, cell lysate before induction; Lane 3 and lane 5 correspond to cell lysate of panel 4 & 6 of **A**, respectively).

E, Native PAGE results show the overexpression and oligomeric state of AcrIIIC1-SS in cell (Lane 2, cell lysate before induction; Lane 3 and lane 5 correspond to cell lysate of panel 4 & 6 of **B**, respectively).

We also described the antimicrobial susceptibility assays in the Methods section, as follows:

“First, using the two-plasmid CRISPR system pCas-pSG, we sought to delete the kanamycin (KAN) gene carried with the pSG plasmid in the industrial E. coli strain BL21 and verified the genome-editing ability by observing a larger inhibition zone of the bacterial lawn of growth, compared to the negative control carrying no spacer gene targeting KAN. To do this, the pCas plasmid was first transformed into the E. coli strain to obtain the pCas-harboring strain, and the cells containing the pCas plasmid were collected and prepared as competent cells. Then, the plasmid pSG-KAN was transformed into the aforementioned pCas-harboring competent cells, and the freshly cultured E. coli cells were suspended, diluted and plated with a kanamycin (100 µg) tablet placed in the center at 30°C overnight. The diameter of the inhibition ring was measured after 20 hrs incubation. A control experiment using bacterial cells harboring pSG plasmid without spacer gene was carried out at the same time. For experimental details, we followed Ji group's publication. To make a quantitative comparison of the size of the inhibition zone, we controlled the amount of the plasmids transformed, the cell turbidity and the culture volume coated, and the incubation time to be precisely consistent among different groups of experiments. Replication of each group of experiment more than three times yields consistent data.

Second, to assess the anti-CRISPR activity of AcrIIIC1, the plasmid expressing WT or mutant AcrIIIC1 was transformed into the pCas-harboring competent cells and prepared as a new kind of competent cell harboring both pCas and pAcrIIIC1 plasmids. Then, the plasmid pSG-KAN or pSG was transformed into the aforementioned pCas-pAcrIIIC1-harboring competent cells and the *E. coli* cells were cultured. The expression of AcrIIIC1 was induced with 0.5 mM of IPTG when the cell density reached an OD₆₀₀ of 0.2. Then as above, when OD₆₀₀ reached 0.6, the cells were suspended, diluted and plated. For assays under oxidizing environments, 1 mM H₂O₂ was added along with IPTG to induce the expression of the oxidized AcrIIIC1 in cell.

To confirm the overexpression of AcrIIIC1 and its redox state, 1mL of culture after IPTG induction for 2~3 hrs was centrifuged, and then the cells were resuspended in 100 µL loading buffer. The AcrIIIC1 expression and its redox state were analyzed on a 12% native PAGE gel.”

Reviewer #2 (Remarks to the Author):

The authors present a structural and biochemical characterization of the redox-dependent interaction of an anti-CRISPR (Acr) protein with NmeCas9. They show, for the first time, that the inhibitory action of AcrIIIC1 is redox-controlled due to a change in oligomeric state that occludes the binding interface with NmeHNH. Using NMR, X-ray crystallography, and BLI, the authors present the structural and binding signatures of the redox-dependent Acr-HNH complex. They extend this work to biochemical assays that corroborate the structural findings, namely that AcrIIIC1 is unable to inhibit NmeCas9 under oxidizing conditions. In general, I find the premise of the work to be strong and the data to be of high quality and rigorously collected. I especially appreciate the internal consistency of several techniques ranging from basic biochemistry to high-level structural biology. This report is novel and has potential to usher in further studies of redox-dependent conformational switching in CRISPR biology, informing us of the role of redox in selective control of Cas9s. There remain several points that should be addressed before the paper can be published.

REPLY: We appreciate the reviewer’s positive assessment and his/her insightful and helpful suggestions provided below, which are very important to improve our manuscript. Accordingly, substantial experiments were performed to address the reviewer’s concerns as detailed below.

1) *The majority of the results presented by the authors describe the “air oxidized” state of AcrIIIC1, rather than a sample treated with oxidant. In several cases, there is clear evidence for only partial oxidation of the critical Cys residues (see Figure 4d, Supplementary Figure 1, Supplementary Figure 4). It is also known from other studies, some of which include mass spectrometry data, that Cys residues may be only partially oxidized under benchtop conditions. While the authors acknowledge this, provide convincing evidence for the redox-driven switch regardless, and conduct some experiments in the presence of peroxide, they should comment (or contrast) the meaning of, and any discrepancies between, the two conditions a bit more.*

REPLY: Thanks for the reviewer's suggestion. We sincerely apologize for the confusion. As for AcrIIIC1 samples used in structural or functional analysis, we checked their redox state with three methods (2D ^1H - ^{15}N NMR spectrum, native PAGE gel, and the Ellman's assay), and confirmed that the samples were in their uniform redox state (either fully reduced or fully oxidized state). However, as the reviewer noticed, under some conditions, a portion of the Cys pair experiences chemical exchange between thiol and disulfide bond due to the intrinsic dynamics as a good redox switch. As for the cases the reviewer mentioned, we'd like to discuss them a bit as follows: For **Fig. 4d** (We didn't label d in the original **Fig. 4**, so we assume the reviewer means the lower panel of **Fig. 4b**), the sample is in its fully oxidized state, supported by the fact that the activity of Cas9 is not inhibited at all even with the excess addition of AcrIIIC1; For **Supplementary Fig. 1a**, the sample is indeed in its mixed states, but the SEC result here is only for a presentation showing there exists two oligomeric states, and this mixture is not used for any further analysis; For the original **Supplementary Fig. 4a** (**Fig. S5a** in the revised manuscript), one pair of Cys17 and Cys80 is indeed in a chemical exchange between thiols and disulfide bond, supported by dual occupancies of the electron density by both thiols and disulfide bond. The sample for crystal growth was confirmed as the fully oxidized state using the methods we mentioned above, but the real buffer for crystal growth that the sample was dissolved in for at least one week, reservoir buffer (1.8 M $(\text{NH}_4)_2\text{SO}_4$, 0.1 M Bis-Tris at pH 6.5 and 2% (v/v) polyethylene glycol monomethyl ether 550) half mixed with AcrIIIC1 storage buffer (20 mM Tris, 100 mM NaCl, pH 7.4), led to a final pH of 6.4, which might favor thiol more than disulfide bond, based on the pKa values measured for the two Cys residues (8.9 for Cys17 and 8.3 for Cys80, respectively).

Following the reviewer's suggestion, we carefully compared the oxidized *NmeAcrIIIC1* proteins induced by the air or the oxidant (H_2O_2) with the methods mentioned above, and confirmed they are identical in terms of oligomeric state, complete loss of free -SH groups, and the structure. During the revision, we repeated the assays involving the oxidized *NmeAcrIIIC1* using an oxidant-induced sample and obtained the same results as before, which is not surprising because we checked and confirmed the molecular state of the samples for every assay in the first place. To make it clear, we added this in the revised manuscript in the section "*NmeAcrIIIC1* assembles into a homodimer triggered by oxidation" in Results, as follows:

*"We further performed native PAGE gel and Ellman's assay to assess the oxidized states of NmeAcrIIIC1 induced in two ways, either by the air or by the oxidant such as H_2O_2 . As shown in **Supplementary Fig. 2a**, the result of native PAGE shows a band with the same size at less than 25 kDa for the oxidized *NmeAcrIIIC1* induced in either way, in comparison with the reduced band at less than 15 kDa. In addition, Ellman's assay result (**Supplementary Fig. 2b**) demonstrates no residual -SH groups for the oxidized *NmeAcrIIIC1* induced in either way, provided an accurate amount of -SH for the reduced *NmeAcrIIIC1* can be calculated against the standard curve fitted with gradient concentrations of DTT mixed with excess DTNB. Collectively, the oxidized forms of *NmeAcrIIIC1* induced in two ways are identical in terms of their oligomeric and redox states. To be consistent, for the following assays, we made the oxidized *NmeAcrIIIC1* by using H_2O_2 , which takes a shorter incubation time as long as an appropriate amount of oxidant is used."*

As mentioned above in the revised manuscript, we do observe some differences between the two processes inducing the oxidized state of AcrIIIC1. First, it takes a much longer time to induce with air, which is reasonable since the oxidizing potential of air is quite moderate. Second, when using oxidant like H₂O₂, it is easy to damage the protein by exposure either to too much amount of oxidant or for too long incubation time, which can be monitored by 2D ¹H-¹⁵N HSQC spectrum showing that the protein is degraded or unfolded.

For the reviewer's convenience, we adapted **Fig. S2b** as **Fig. R2-1**:

Figure R2-1 Characterization of the oxidized *NmeAcrIIIC1* induced in two ways.

A, Native PAGE gel shows the bands of the reduced *NmeAcrIIIC1* (Lane 1), the oxidized *NmeAcrIIIC1* induced by the air (Lane 2), and the oxidized *NmeAcrIIIC1* induced with H₂O₂ (Lane 3).

B, Standard curve for DTT+DTNB mixture from Ellman's assays. The number of free sulfhydryl groups for the reduced *NmeAcrIIIC1* can be accurately extracted from the curve, while no absorbance at 412 nm can be detected for either form of oxidized *NmeAcrIIIC1*.

C, Overlay of ¹H-¹⁵N HSQCs for both forms of oxidized *NmeAcrIIIC1*.

2) On a somewhat related note, there is no mention of redox potentials accessed during the course of these experiments, or how these conditions would compare to those found in a cell. Some swings in biological redox potentials are quantified in the literature, as is their effect on protein stability and function. If there is any context for the experimental conditions reported here that can be discussed based on this knowledge, it would be a welcome addition.

REPLY: Thanks for the reviewer's insightful suggestion. Following the reviewer's suggestion, we made extensive efforts on measuring the redox potentials for all the experimental conditions we used. For the buffers with a dominating reductant such as DTT or TCEP, we were able to obtain a relatively stable redox potential value, as listed in **Supplementary Table 4** in the revised manuscript. However, for conditions without reductants where the dissolved oxygen seems to dominate the redox potential (the

highest concentration of H₂O₂ we used in assays was 0.02 mM, which can't compete out the dissolved oxygen and stabilize the redox potential), we constantly observed a varied redox potential ranging from -50 mV to 300 mV, which also drops significantly (> 50 mV per day) probably due to the loss of the dissolved oxygen along time. Therefore, we included a few sentences discussing the redox potentials measured for reducing buffers in comparison with normal cellular redox potential reported previously, as follows:

*“To assess the intracellular state of AcrIIIC1, we measured the redox potentials for the reducing buffers we used (listed as **Supplementary Table 4**), and compared them with the intracellular redox potentials reported previously. It seems that normal cytoplasmic values (-260 mV to -200 mV) fall within the range of our reducing buffers (-360 mV to -200 mV), indicating most of the AcrIIIC1 molecules should be at their reduced state under normal physiological conditions. Under oxidative stress, local redox potential elevation in cell is not easy to measure at this point, due to the lack of a stable redox sensor working in this condition. However, it was reported that cells exposed to a more oxidized potential (0 mV) had stimulated H₂O₂ production, which can induce the inactive dimer of AcrIIIC1 based on our in vitro and in vivo data (**Supplementary Fig. 7**).”*

For the reviewer's convenience, we paste the newly added **Supplementary Table 4** here, as below:

Supplementary Table 4: Redox potentials measured for the reducing buffers.

Reaction buffers	Redox potential vs. Ag/AgCl (mV)	Calibrated potential vs. SHE (Eh) (mV)
20 mM HEPES, 100 mM KCl, 10 mM MgCl ₂ , and 5% (w/v) glycerol pH 7.5, 5 mM DTT	-560 ± 34	-360 ± 34
20 mM Tris, 100 mM NaCl, pH 7.4, 1 mM DTT	-400 ± 38	-200 ± 38

3) Page 7, lines 124-128 – the authors discuss the lack of inhibition by air oxidized NmeAcrIIIC1 toward NmeCas9. Has this effect been observed in the literature for any other Cas9 species? It seems reasonable to assume that if this manuscript is the first report of redox-switching governing the efficacy of Acr proteins, then prior biochemical assays of Cas9 inhibition by Acrs may not have controlled for redox state. Does this mean that some prior assays could be clouded by the uncertainty in the redox conditions? If not, why not? There are moments when the authors discuss how wide-ranging such a mechanism might be for Acr proteins, but this seems like an important point of context here.

REPLY: Thanks for the reviewer's valuable thoughts and comments. As a matter of fact, reviewer #1 has a similar concern as well. To answer the first question “Has this effect been observed in the literature for any other Cas9 species?” We carefully surveyed the

previous publications, the answer is NO. We think the possible reason could be that, as for protein purification for structural biology studies or biochemical assays, excess DTT or TCEP is routinely added to prevent potential mismatched intermolecular disulfide bond linkage of the target protein, especially for proteins without prior knowledge of redox regulation involved. To support this, we did a survey of literature with *in vitro* AcrIIIC1 assays involved, and found that all of the assays were carried out under reducing conditions with different kinds of reductants added in the buffer, listed as **Supplementary Table 3**. Therefore, in this way, prior *in vitro* assays did have good control of the reduced state of AcrIIIC1. As for the assays *in vivo*, we think at least some prior assays could be clouded by the uncertainty in the redox conditions, which is also one of the reasons why we generated the AcrIIIC1-SS mutant. Accordingly, we added discussion regarding this question on page 14 in the revised manuscript as follows:

*“One may wonder why previous studies haven’t identified the oxidized form of NmeAcrIIIC1. After surveying 9 publications that investigated AcrIIIC1 experimentally, including structural studies, in vitro DNA cleavage or binding assays, we find that previous in vitro assays were all performed with reductants maintaining reducing experimental conditions (listed as **Supplementary Table 3**). For example, in the structural study of NmeAcrIIIC1 complexed with HNH by Doudna group in 2017, they kept 0.5 -1 mM TCEP-NaOH in all purification buffers and 1 mM DTT or 1mM TCEP-NaOH for all in vitro cleavage and binding assays. It could be the reducing experimental condition that precluded the chance of forming the oxidized AcrIIIC1. As for previous in vivo assays, since only apparent activity of AcrIIIC1 can be observed without characterization of the molecular state, it’s hard to tell the exact redox state of AcrIIIC1 molecules in cell. To assess the intracellular state of AcrIIIC1, we measured the redox potentials for the reducing buffers we used (listed as **Supplementary Table 4**), and compared them with the intracellular redox potentials reported previously. It seems that normal cytoplasmic values (-260 mV to -200 mV) fall within the range of our reducing buffers (-360 mV to -200 mV), indicating most of the AcrIIIC1 molecules should be at their reduced state under normal physiological conditions. Under oxidative stress, local redox potential elevation in cell is not easy to measure at this point, due to the lack of a stable redox sensor working in this condition. However, it was reported that cells exposed to a more oxidized potential (0 mV) had stimulated H₂O₂ production, which can induce the inactive dimer of AcrIIIC1 based on our in vitro and in vivo data (**Supplementary Fig. 7**).”*

For the reviewer’s convenience, we paste the newly added **Supplementary Table 3** here, as below:

Supplementary Table 3: Reducing buffer conditions used in previous AcrIIc1 studies *in vitro*.

Publications	Protein purification	In vitro DNA cleavage assay	Binding assay with Cas9
Mathony, J. et al. Computational design of anti-CRISPR proteins with improved inhibition potency. Nat. Chem. Biol. 16,725-730(2020).	—	20 mM Tris, pH 7.5, 100 mM KCl, 5 mM MgCl ₂ , 5% glycerol, 1 mM DTT	—
Kim, Y. et al. Anti-CRISPR AcrIIc3 discriminates between Cas9 orthologs via targeting the variable surface of the HNH nuclease domain. FEBS J. 286, 4661-4674(2019).	—	20 mM sodium phosphate, pH 6.5, 50 mM NaCl, 1 mM PMSF 2 mM BME	20 mM sodium phosphate, pH 7.4, 100 mM NaCl, 5 mM BME
Zhu, Y. et al. Diverse Mechanisms of CRISPR-Cas9 Inhibition by Type IIC Anti-CRISPR Proteins. Mol. Cell 74, 296-309.e7(2019).	20 mM Tris-HCl, pH 7.5, 300 mM NaCl, 2 mM MgCl ₂ , 1 mM DTT	20 mM Tris-HCl, pH 7.5, 100 mM KCl, 5 mM MgCl ₂ , 5% glycerol, 5 mM DTT	20 mM Tris-HCl, pH 7.5, 300 mM NaCl, 2 mM MgCl ₂ , 1 mM DTT
Phaneuf, CR. et al. Ultrasensitive Multi-Species Detection of CRISPR-Cas9 by a Portable Centrifugal Microfluidic Platform. Anal Methods. 11, 559-565(2019).	50 mM Tris-HCl, pH 7.5, 500 mM NaCl, 10% glycerol, 0.5 mM TCEP	—	—
Edraki, A. et al. A Compact, High-Accuracy Cas9 with a Dinucleotide PAM for In Vivo Genome Editing. Mol Cell. 73, 714-726.e4(2019).	50 mM Tris-HCl, pH 7.5, 500 mM NaCl, 5 mM imidazole, 1 mM DTT	—	—
Seamon, KJ. et al. Versatile High-Throughput Fluorescence Assay for Monitoring Cas9 Activity. Anal Chem. 90, 6913-6921(2018).	50 mM Tris-HCl, pH 7.5, 100 mM NaCl, 0.5 mM TCEP	—	—
Rousseau, BA. et al. Programmable RNA Cleavage and Recognition by a Natural CRISPR-Cas9 System from Neisseria meningitidis. Mol Cell. 69,906-914.e4(2018).	20 mM HEPES, pH 7.5, 300 mM NaCl, 0.5 mM DTT	20 mM HEPES, pH 7.5, 50 mM KCl, 0.1 mM EDTA, 10 mM MgCl ₂ , 0.5 mM DTT	—
Harrington, L. B. et al. A Broad-Spectrum Inhibitor of CRISPR-Cas9. Cell 170, 1224-1233.e15 (2017).	50 mM Tris-HCl, pH 7.5, 20 mM imidazole, 500 mM NaCl, 1 mM PMSF, 0.5 mM TCEP-NaOH	20 mM Tris-HCl, pH 7.5, 100 mM KCl, 5 mM MgCl ₂ , 5% glycerol, 1 mM DTT	20 mM HEPES-NaOH, pH 7.5, 150 mM NaCl, 1 mM TCEP-NaOH
Pawluk, A. et al. Naturally Occurring Off-Switches for CRISPR-Cas9. Cell 167, 1829-1838.e9(2016).	10 mM Tris, pH 7.5, 250mM NaCl, 5mM BME	20 mM HEPES-KOH, pH 7.5, 150 mM KCl, 10% glycerol, 10 mM MgCl ₂ , 1 mM DTT	—

4) It is intriguing that both the monomeric (reduced) and dimeric (oxidized) forms of AcrIIIC1 are rigid structures (via NMR relaxation) when it is widely accepted that Cas9 proteins themselves are highly dynamic. It is possible the motion exists on other timescales, but it might be worth mentioning how/if this Acr is proposed to hinder dynamics of the system when interacting.

REPLY: This is a very good point, thanks for the reviewer's suggestion. We totally agree with the reviewer that dynamics and/or stability of the components in the CRISPR system play a crucial role in the catalytic activity, and very possibly Acrs hinder the dynamics of the system when interacting. As for AcrIIIC1, a previous study by Niopek group⁷ has demonstrated that chimeric AcrIIIC1s containing protein domains inserted into the distal site from the HNH-interacting surface enhanced its potency of Cas9 inhibition significantly, provided the interaction between HNH and AcrIIIC1 is already tight enough (1~6 nM of K_d), implying the chimeras must hamper the conformational rearrangement of the Cas9 subunits from assembling its catalytically active state somehow. Following the reviewer's suggestion, we proposed a model (**Fig. R2-4**) demonstrating how different redox states of AcrIIIC1 would affect the dynamics of the Cas9 subunits and regulate its activity, in the very last paragraph of the Discussion in the revised manuscript. The text is as follows:

*“Utilizing the *Sau*Cas9 complex structure with its HNH domain bound to AcrIIA14, we propose a model demonstrating how different redox states of AcrIIIC1 would affect the dynamics of the Cas9 subunits and regulate Cas9's activity. When sensing the environmental redox potential elevation, AcrIIIC1 molecule can dimerize with its closest partner and then loses its contact interface with HNH, so HNH may complete its conformational rearrangement positioning its catalytic site onto the DNA substrate. In contrast, in the reduced cellular environment, the monomeric AcrIIIC1 can bind to HNH and restrain the Cas9 complex at a precatalytic conformation without activity.”*

We added the model as **Fig. S9**, adapted as **Figure R2-4** shown as below:

Figure R2-4 Schematic model demonstrating how different redox states of AcrIIIC1 affects the dynamics of Cas9 subunits and Cas9's activity.

5) Page 10, line 219 – the term “design” here seems like a bit of an overstatement, since the two Cys residues are native to the Acr protein and this study is simply removing them. It is very strong proof that these residues are involved in the redox switching, but in my opinion, “design” implies more than basic mutagenesis. Perhaps if Cys residues were inserted at another point within the protein with some measure of the same functionality, it would apply.

REPLY: Thanks for the reviewer’s suggestion. We agree that the word “design” here is an overstatement. Following the reviewer’s suggestion, we changed “design” as well as “engineer” to “generate”, and highlighted them in red in the revised manuscript.

6) Figure 1, panel a – the authors should show the two chromatograms overlaid on top of each other to better demonstrate the peak shift. Were any experiments with molecular weight standards carried out by SEC?

REPLY: Following the reviewer's suggestion, we overlaid the gel filtration chromatograms of two redox states of AcrIIIC1 and replaced **Fig. 1a** as below (**Fig. R2-6A**). In addition, we supplemented the SEC calibration curve calibrated with three globular protein standards as **Fig. S1b**, adapted as **Fig. R2-6B** as below.

Figure R2-6 The MW of the two redox states of *NmeAcrIIIC1* calculated by the SEC calibration curve.

A, SEC runs show that the retention volume of *NmeAcrIIIC1* changes under different redox conditions. When oxidized, *NmeAcrIIIC1*'s apparent molecular weight becomes twice as the reduced one, corresponding to a dimer.

B, The calibration curve for our SEC column (Superdex 75 Increase 10/300 GL (Cytiva)) using three globular protein standards.

7) Figure 2 – panels A and B seem like they should be swapped.

REPLY: Thanks for the reviewer's suggestion. In the revised manuscript, we swapped panels A and B of **Fig. 2**.

8) Figure 2 – on a related note in panel b, are the traces concentration normalized? This information might be useful in the caption to more easily interpret the shifts (or lack thereof) in the red traces corresponding to the complexes.

REPLY: Yes, the concentration was normalized to 0.1 mM of a single protein for each trace. Following the reviewer's suggestion, we revised the figure caption accordingly.

9) Figure 4, panel b – this spectrum should be compared to an oxidized spectrum, similarly to the comparison in panel a.

REPLY: Thanks for the reviewer’s suggestion. In the revised manuscript, we replaced Fig. 4b with overlaid HSQCs, as suggested by the reviewer. Figure R2-9 is adapted from the new Fig. 4b:

Figure R2-9 Superimposition of ^1H - ^{15}N HSQCs of the oxidized *NmeAcrIIC1* before (black) and after (blue) recycling from the reduced state.

10) Figure 5, panel b – it is not clear from the caption which conditions correspond to the BLI plot. In the “SS” mutant, the authors note binding under both redox conditions, due to the removal of Cys and prevention of the dimer. However, only one set of traces appears and the redox state is not indicated. Do data exist for both redox states?

REPLY: We apologize for the confusion. The original BLI data was collected under the reducing condition, as mentioned in the manuscript. In the revised manuscript, we supplemented the BLI data collected under the non-reducing condition as Fig. S6a, and revised the text as follows:

“With or without reductants in the buffer, *AcrIIC1*-SS exhibits a K_d to *NmeHnh* at 12.7 or 10.8 nM (Fig.5b and Supplementary Fig. 6a), achieving comparable binding affinity as that of the wild-type *AcrIIC1* (wt*AcrIIC1*).”

For the reviewer’s convenience, we adapted Fig. 5b and Fig. S6a as Fig. R2-10a,b as below:

Figure R2-10 K_d determined by BLI for AcrIIIC1-SS bound to *NmeHNH* under different redox conditions.

A, With reductant in the buffer, K_d determined by BLI for AcrIIIC1-SS bound to *NmeHNH* is 12.7 nM.

B, Without reductant in the buffer, K_d determined by BLI for AcrIIIC1-SS bound to *NmeHNH* is 10.8 nM.

11) *Supplementary Figure 4 – it is hard to imagine such a small error (± 0.054) in the pK_a determination of Cys17 when the pH plot has not yet begun to level off in the high pH regime.*

REPLY: We apologize for the confusion. We understand the reviewer’s concern that for Cys17 with a high pK_a , using a limited set of data points ($pH < 9$) to fit the curve might yield a biased pK_a value. To address this issue, we re-analyzed the spectra, and confirmed that the chemical shift for C_β of Cys17 at $pH 9.5$ is too weak to be measured precisely. However, we do find that our replicated data was collected from the same batch of NMR samples at a very close time, so the error for each data point was very small. Therefore, we re-collected the data with a different batch of NMR sample, and re-fit the curve as **Fig. S5b**. The new curve results in a very close pK_a value of 8.92 (compared to the old 8.90) for Cys17, which is not surprising because chemical shifts for a stable protein sample don’t vary much provided with a good calibration with the standards. The error is bigger but still relatively small (at ± 0.158 this time). The error shown here only represents the deviation of the pK_a value one would expect, based on the best curve fitting to the limited experimental data we collected for the modified Henderson-Hasselbalch equation we input (**Eq. 2**). And we agree the calculated pK_a value for Cys17 might be biased.

12) There are many grammatical and typographic errors throughout the manuscript. Recommend carefully reading through again to correct these issues.

REPLY: We appreciate the reviewer's careful reading. Following the reviewer's suggestion, we read through the manuscript very carefully and corrected all of the grammatical and typographic errors we could identify, which are labeled in red in the revised manuscript.

References

1. Martin, O. A., Villegas, M. E., Vila, J. A. & Scheraga, H. A. Analysis of $^{13}\text{C}\alpha$ and $^{13}\text{C}\beta$ chemical shifts of cysteine and cystine residues in proteins: a quantum chemical approach. *J. Biomol. NMR* **46**, 217–225 (2010).
2. Zhu, Y. *et al.* Diverse Mechanisms of CRISPR-Cas9 Inhibition by Type IIC Anti-CRISPR Proteins. *Mol. Cell* **74**, 296-309.e7 (2019).
3. Stone, N. P. *et al.* A Hyperthermophilic Phage Decoration Protein Suggests Common Evolutionary Origin with Herpesvirus Triplex Proteins and an Anti-CRISPR Protein. *Structure* **26**, 936-947.e3 (2018).
4. Phaneuf, C. R. *et al.* Ultrasensitive multi-species detection of CRISPR-Cas9 by a portable centrifugal microfluidic platform. *Anal. Methods* **11**, 559–565 (2019).
5. Seamon, K. J., Light, Y. K., Saada, E. A., Schoeniger, J. S. & Harmon, B. Versatile High-Throughput Fluorescence Assay for Monitoring Cas9 Activity. *Anal. Chem.* **90**, 6913–6921 (2018).
6. Harrington, L. B. *et al.* A Broad-Spectrum Inhibitor of CRISPR-Cas9. *Cell* **170**, 1224-1233.e15 (2017).
7. Mathony, J. *et al.* Computational design of anti-CRISPR proteins with improved inhibition potency. *Nat. Chem. Biol.* **16**, 725–730 (2020).
8. Kim, Y. *et al.* Anti-CRISPR AcrIIIC3 discriminates between Cas9 orthologs via targeting the variable surface of the HNH nuclease domain. *FEBS J.* **286**, 4661–4674 (2019).
9. Pawluk, A. *et al.* Naturally Occurring Off-Switches for CRISPR-Cas9. *Cell* **167**, 1829-1838.e9 (2016).
10. Rousseau, B. A., Hou, Z., Gramelspacher, M. J. & Zhang, Y. Programmable RNA Cleavage and Recognition by a Natural CRISPR-Cas9 System from *Neisseria meningitidis*. *Mol. Cell* **69**, 906-914.e4 (2018).
11. Edraki, A. *et al.* A Compact, High-Accuracy Cas9 with a Dinucleotide PAM for In Vivo Genome Editing. *Mol. Cell* **73**, 714-726.e4 (2019).

Point-by-point responses to the reviewers' comments:

* The reviewers' comments are in *Italic*. Author's responses are in blue. All revisions are highlighted in red in the revised manuscript.

Reviewer #1 (Remarks to the Author):

*They tried to improve this manuscript with additional supporting experiments.
Most of my curiosities were solved in this moment.*

My final suggestion is to move S7 (antimicrobial assay) and S9 (Suggested model) to main text. These two supple figures are very informative.

REPLY: We sincerely thank this reviewer for his/her positive assessment. Following the reviewer's suggestions, we moved Supplementary Figure 7 panels a,b,c to the main text as Figure 6, and changed panels d,e of Supplementary Figure 7 into a,b. The reason why we kept panels d,e in S7 is that we think panels a,b,c are sufficient to show the result, and we wish to keep the main figures to be informative. In addition, we moved Supplementary Figure 9 to the main text as Figure 7.

Reviewer #2 (Remarks to the Author):

In the revised version of this manuscript, the authors have done an exceptional job of carefully addressing the concerns of the original review. I feel with these changes, particularly some added context and figure clarifications, the manuscript can now be accepted for publication.

REPLY: We sincerely thank this reviewer for his/her positive assessment.